# Replacing the PDZ-interacting C-termini of DSCAM and DSCAML1 with epitope tags causes different phenotypic severity in different cell populations

Andrew M Garrett[1], Abigail LD Tadenev[1], Yuna T Hammond[1], Peter G Fuerst[2,3], Robert W Burgess[1]*

[1]The Jackson Laboratory, Bar Harbor, United States; [2]Department of Biological Sciences, University of Idaho, Moscow, United States; [3]WWAMI Medical Education Program, University of Idaho, Moscow, United States

**Abstract** Different types of neurons in the retina are organized vertically into layers and horizontally in a mosaic pattern that helps ensure proper neural network formation and information processing throughout the visual field. The vertebrate Dscams (DSCAM and DSCAML1) are cell adhesion molecules that support the development of this organization by promoting self-avoidance at the level of cell types, promoting normal developmental cell death, and directing vertical neurite stratification. To understand the molecular interactions required for these activities, we tested the functional significance of the interaction between the C-terminus of the Dscams and multi-PDZ domain-containing scaffolding proteins in mouse. We hypothesized that this PDZ-interacting domain would mediate a subset of the Dscams' functions. Instead, we found that in the absence of these interactions, some cell types developed almost normally, while others resembled complete loss of function. Thus, we show differential dependence on this domain for Dscams' functions in different cell types.

*For correspondence: robert.
burgess@jax.org

**Competing interests:** The authors declare that no competing interests exist.

## Introduction

The vertebrate retina provides an advantageous model to study how specific neuronal cell types organize themselves during development to form functional circuits. The ~100 neuronal cell types of the retina vertically organize into layers, with light-transducing photoreceptors in the outermost cellular layer (ONL, or outer nuclear layer) and retinal ganglion cells (RGCs) – responsible for the sole output from the retina – in the innermost layer (RGL, or retinal ganglion cell layer). Between these, both functionally and physically, are the interneuron cell types (horizontal, amacrine, and bipolar cells) in the inner nuclear layer (INL), which process and relay visual information from photoreceptors to RGCs. These three cellular layers are separated by two synaptic plexiform layers, where the neurites from these cell types mingle and form synaptic connections with specific partners in specific substrata. Additionally, many cell types are horizontally spaced in a mosaic pattern, such that there is a low probability of finding two neurons of the same subtype (i.e., homotypic) in close proximity. This pattern ensures that the information processing provided by each subtype is distributed across the retina (*Masland, 2012*).

Establishing this pattern requires cells to be able to recognize other cells of the same type and avoid them, while also stably interacting with appropriate synaptic partners (*Garrett and Burgess, 2011*). Critical parts of this recognition code include Down syndrome cell adhesion molecule (DSCAM) and the highly similar Dscam-like1 (DSCAML1), collectively referred to as Dscams. *Dscam*

**eLife digest** Neurons in a part of the eye called the retina detect light and convert it into electrical signals that are sent to the brain. Different types of neurons in the retina are arranged vertically into layers and horizontally in a mosaic pattern so that two neurons of the same type are not next to each other. To establish this highly organized pattern, neurons in the developing retina must be able to recognize other neurons of the same type and avoid moving towards them – a process referred to as self-avoidance.

A group of proteins called the Dscams are found on the surface of neurons and play key roles in positioning them in the retina. Dscams promote self-avoidance, help to establish connections between certain neurons and kill any excess neurons that are not needed. However, the mechanisms by which Dscams serve these three roles were not known.

Scaffolding proteins in the cell interior interact with Dscams to hold them in place on the cell surface. Garrett et al. investigated whether Dscams need to interact with the scaffolding proteins in order to carry out any of their activities.

The experiments used mice that had been genetically engineered to produce mutant Dscam proteins that cannot bind to the scaffolding proteins. Garrett et al. hypothesized that this would affect the activities of Dscams in all of the different types of neurons in the retina. However, the experiments show that the mutant Dscam proteins had different effects on the neurons. Some types of neurons developed normally, while others experienced disruptions in all three of the processes that Dscams are normally involved in. Some other neurons were affected to a moderate extent. This indicates that Dscams use different mechanisms in different types of neurons to carry out the same activities. The next step is to find out what other proteins Dscams need to interact with in different types of neurons.

and *Dscaml1* encode homophilic members of the Ig-superfamily of cell adhesion molecules, and are expressed in non-overlapping neuronal subtypes in the retina (*Agarwala et al., 2001*; *Fuerst et al., 2009*; *Yamagata and Sanes, 2008*). The Dscams promote self-avoidance at the cell type level: When either gene is mutated, the cell types that normally express the Dscam lose their mosaic spacing and often form clusters (*Fuerst et al., 2009*, *2008*). The neurites fail to evenly cover their receptive fields, and instead form fascicles with neighboring homotypic cells. This clustering and fasciculation is, with few exceptions, homotypic – cells of one subtype rarely cluster with the cells of another subtype. Self-avoidance requires homophilic Dscam interactions between cells, demonstrated in mosaic experiments where neurons lacking *Dscam* fasciculate with homotypic neurons with intact *Dscam* (*Fuerst et al., 2012*). This self-avoidance function is consistent with studies in *Drosophila*, which have four Dscam genes. Most notably, *Dscam1* promotes self-avoidance at the individual cell level by using alternative splicing to produce 19,008 distinctly homophilic isoforms, allowing each neuron to recognize and avoid 'self' while still interacting with 'non-self' during processes like dendrite arborization and axon branching (reviewed in [*Zipursky and Grueber, 2013*]).

Dscams are also required for normal developmental cell death. In the mutant mice, there is an overabundance of each affected cell type, resulting in a severe expansion of the retina through the cellular and plexiform layers. The extent of cell number expansion varies with cell type. Some cell types are expanded beyond even that seen in mutants for the pro-apoptotic *Bax* gene, while others are more modestly expanded compared to *Bax* mutants (*Fuerst et al., 2009*, *2008*; *Keeley et al., 2012*).

Dscams also contribute to the vertical organization of the retina. In chick, Dscams label-specific sublaminae of the IPL and can instruct neurite targeting to these layers (*Yamagata and Sanes, 2008*). DSCAM protein localization is punctate throughout the IPL in mouse, and is not confined to specific sublaminae (*de Andrade et al., 2014*). Despite this, some neuronal types do have disorganized neurite stratification in *Dscam* mutants, although the disorganization varies with genetic background (*Fuerst et al., 2010*). There are also indications that the synaptic connections that form do not mature normally. For instance, *Dscaml1* is expressed both in rod bipolar cells and AII amacrine cells, which connect at dyad ribbon synapses in the IPL. These synapses can still be found in *Dscaml1*

mutants, but are morphologically abnormal, with indistinct, detached presynaptic ribbons, and functionally abnormal, with much slower decay of the synaptic current (*Fuerst et al., 2009*).

The mechanisms by which the Dscams mediate their developmental functions are unknown. One attractive hypothesis is that different functions, including cell death, self-avoidance, and synapse maturation, are mediated by different molecular interactions in the cytosol. Signaling molecules have been found in complex with Dscam1 in *Drosophila* (e.g., Dock/Pak, Ableson, tubulin binding cofactor D [*Okumura et al., 2015*; *Schmucker et al., 2000*; *Sterne et al., 2015*]) and DSCAM in vertebrates (e.g., PAK1, FAK, Fyn [*Purohit et al., 2012*]). Both DSCAM and DSCAML1 also have canonical PDZ-interacting motifs at their C-termini by which they interact with scaffolding proteins in the MAGI (membrane-associated guanylate kinase with inverted orientation) and PSD95 families (*Yamagata and Sanes, 2010*). Because this motif is common to both Dscams, we chose to test the functional significance of these C-terminal interactions, with the initial hypothesis that this interaction would be required for a specific subset of Dscams' functions.

We engineered mouse mutations in which the sequences encoding the final 10 amino acids of DSCAM and DSCAML1, which interact with PDZ domain-containing proteins, were replaced with epitope tags. Contrary to our initial hypothesis, we found that rather than distinguishing phenotypes based on specific molecular mechanisms, these mutations distinguished cell types. Some cell types were essentially like controls in cell number, spacing, and stratification, whereas others were nearly as disrupted as null mutants. Still other cell types were intermediate in severity. Together, these results demonstrate that different cell types have different dependencies on the PDZ-interacting C-termini of Dscams for function, indicating multiple intracellular molecular mechanisms are involved.

## Results

### Deletion of the PDZ-interacting C-terminus of DSCAM

The molecular interactions through which DSCAM could function may involve extracellular interactions, intracellular interactions with other membrane proteins, or initiation of intracellular signaling pathways. We reasoned that a tractable first step in dissecting these possibilities would be to disrupt the C-terminus of DSCAM. Both DSCAM and DSCAML1 have canonical PDZ-interacting domains at their C-termini, and have been shown to interact with PSD-95 and MAGI family members (*Yamagata and Sanes, 2010*) (*Figure 1—figure supplement 1*).

To assess the functional relevance of DSCAM's PDZ-binding motif, we generated a targeted allele of *Dscam*, in which the sequence encoding the C-terminal 10 amino acids was replaced with sequence encoding a Myc epitope tag (*Dscam$^{\Delta C}$*, see *Figure 1A* and Materials and methods). Deletion of these final 10 amino acids disrupts the canonical binding to the hydrophobic pocket of PDZ domains (*Doyle et al., 1996*), and this DSCAM-ΔC mutation markedly reduced MAGI-3 association in co-immunoprecipitation experiments with the DSCAM intracellular domain (ICD) when co-transfected in HEK293T cells (*Figure 1B*). The residual interaction may be an artifact of overexpression, or may reflect interactions between the ICD of DSCAM and MAGI3 that are not dependent on the canonical C-terminal PDZ-interacting domain, but nonetheless, the affinity is greatly reduced and given results described below for DSCAML1-△C, we have successfully interfered with the interactions between Dscams and PDZ-domain-containing proteins using this strategy. Unlike null mutants, which do not survive on a C57BL/6 background, *Dscam$^{\Delta C/\Delta C}$* mice survived without hydrocephaly or any of the overt phenotypes observed in null animals (*Amano et al., 2009*; *Fuerst et al., 2010*, *2008*).

Consistent with the milder phenotype of *Dscam$^{\Delta C/\Delta C}$* mice, the protein appears to be stable and properly localized. When transfected into HEK293T cells, DSCAM-ΔC protein was produced and targeted to the membrane at levels similar to that of full length DSCAM (*Figure 1—figure supplement 1*). To test if DSCAM-ΔC protein was stable in vivo, we immunoprecipitated protein from neonatal brains using an antibody against DSCAM, and performed Western blots with a second anti-DSCAM antibody. The relative abundance and size of DSCAM protein was indistinguishable between *Dscam$^{+/+}$* and *Dscam$^{\Delta C/\Delta C}$* samples, while negative control *Dscam$^{-/-}$* samples were devoid of DSCAM protein, as expected (*Figure 1C*). In the retina, DSCAM acquires a punctate localization in the synaptic plexiform layers (*de Andrade et al., 2014*). At three weeks of age, there was no obvious

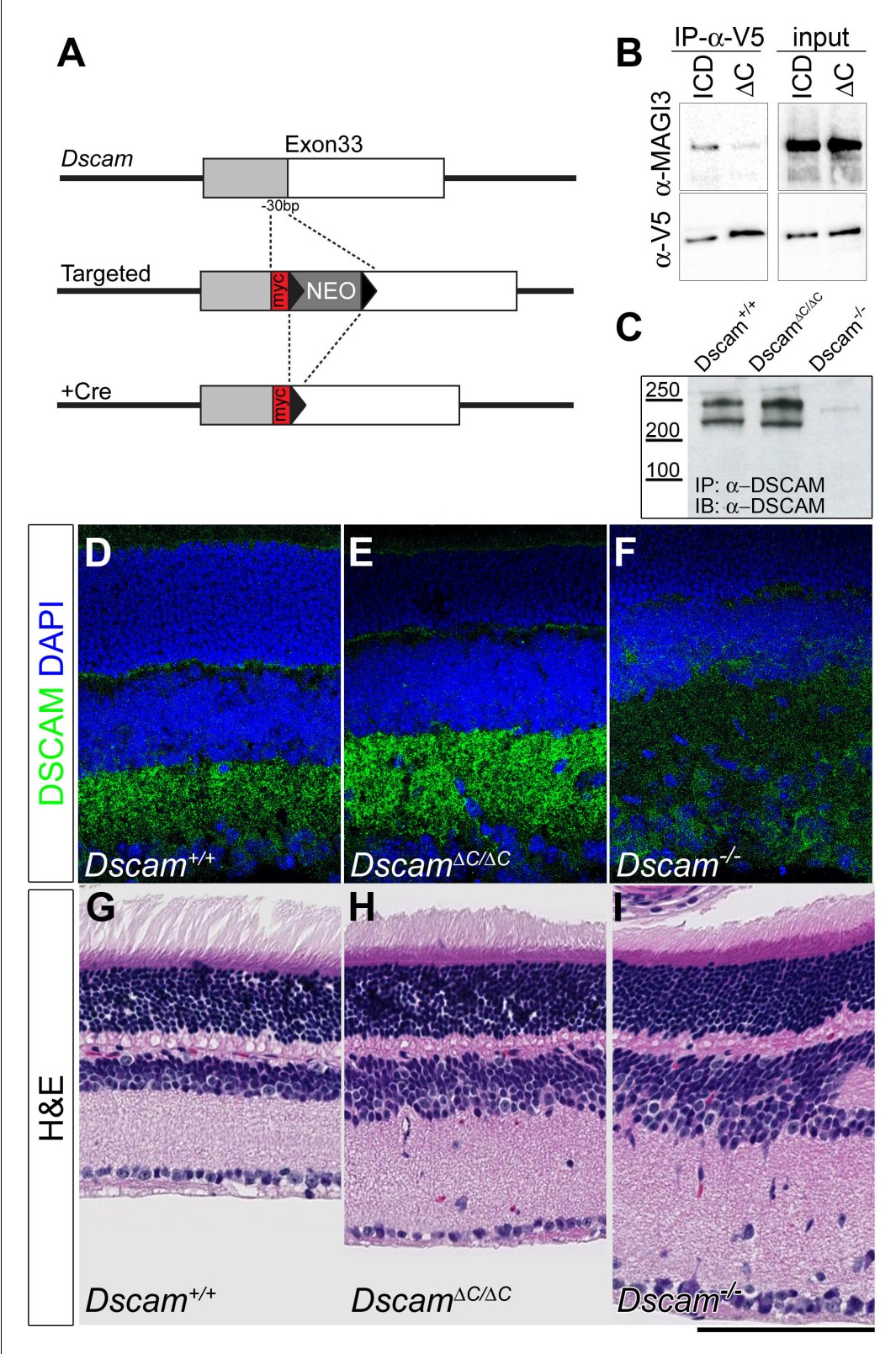

**Figure 1.** The C-terminus of DSCAM is not required for protein stability or localization. (A) In the *Dscam*$^{\Delta C}$ allele, the sequence encoding the final ten amino acids was replaced with a Myc tag by homologous recombination. (B) Western blots of protein immunoprecipitated from HEK293T cells co-transfected with MAGI-3 and V5-tagged DSCAM intracellular domain (ICD) or V5-tagged DSCAM-ΔC ICD (ΔC) demonstrates that the ΔC mutation disrupts the PDZ-binding of DSCAM's C-terminus. (C) Western blots of DSCAM protein immunoprecipitated from neonatal brains showed no change in

*Figure 1 continued on next page*

*Figure 1 continued*
the size or amount of DSCAM in *Dscam*^ΔC/ΔC^ mutants. The antibody specificity is confirmed by the lack of signal from *Dscam*^-/-^ brains. (**D–F**) Immunofluorescent labeling for DSCAM in vertically sectioned retinas from 3-week old mice demonstrated that the protein is found in a normal, punctate localization in the synaptic plexiform layers in *Dscam*^+/+^ (**D**) and *Dscam*^ΔC/ΔC^ (**E**) mice, consistent with earlier reports for wild type DSCAM (*de Andrade et al., 2014*). No staining above background was found in *Dscam*^-/-^ retinas (**F**), demonstrating the specificity of the DSCAM antibody. (**G–I**) Hematoxylin and eosin staining of adult retinas shows that, compared to controls (**G**), *Dscam*^-/-^ retinas (**I**) are severely expanded and disorganized. *Dscam*^ΔC/ΔC^ retinas (**H**) have modest expansion, but not the extensive disorganization found in the null mutant. Scale bar is 100 µm. See also *Figure 1—figure supplement 1* and *Figure 1—figure supplement 2*.

The following figure supplements are available for figure 1:

**Figure supplement 1.** DSCAM's C-terminus interacts with PDZ domains.

**Figure supplement 2.** DSCAM protein localization is grossly unchanged in *Dscam*^ΔC/ΔC^ retinas.

mislocalization or reduction in labeling in cryosections from *Dscam*^ΔC/ΔC^ retinas labeled for DSCAM by immunofluorescence (*Figure 1D–F*, and *Figure 1—figure supplement 2*).

In contrast to spontaneous *Dscam*^-/-^ mutants, which display severely disrupted retinal histology, with expanded and grossly disorganized inner nuclear, inner plexiform, and retinal ganglion cell layers (*Fuerst et al., 2008*) (*Figure 1G,I*), *Dscam*^ΔC/ΔC^ retinas showed some expansion and disorganization, but a milder phenotype than in *Dscam*^-/-^ animals (*Figure 1H*). Thus, *Dscam*^ΔC/ΔC^ mice produce and localize DSCAM normally, and have a less severe gross histological defect. This intermediate phenotype could be the result of milder, partial-loss-of-function phenotypes in all *Dscam*-expressing cell types, or it could be that now only some cell types display the *Dscam* phenotype, whereas others are normal. The latter appears to be the case, based on our results below.

## Differential dependence DSCAM/PDZ interactions across cell types

We analyzed the spacing and density of dopaminergic amacrine cells (DA cells, tyrosine hydroxylase-positive, *Figure 2A–C*) and bNOS-positive amacrine cells (*Figure 2H–J*), both of which form homotypic clusters and increase in number in *Dscam*^-/-^ mutants (*Fuerst et al., 2008*). We measured cell spacing with three distinct tests: density recovery profiling (DRP), Voronoi tessellation domain analysis, and nearest neighbor analysis. Each of these tests provides a measure of spacing independent of overall density. DRP plots the density of cells at binned distances from each individual cell. When cells are randomly spaced, the density within each bin is equal to the overall density. When cells are mosaically spaced, there is an 'exclusion zone' where bins close to each reference cell have a lower density than the overall field. If cells are clustered then the near bins have a higher density than the overall field (*Rodieck, 1991*). In addition to providing a visual representation of cell spacing, DRP also calculates a 'packing factor,' which we used for statistical comparison. A perfectly ordered array of cells has a packing factor of 1, while an array of cells with no exclusion zone has a packing factor of 0 (*Rodieck, 1991*). In Voronoi tessellation domain analysis, each point in the image is assigned to the domain of the nearest cell, creating a tessellation (e.g., *Figure 2—figure supplement 1*). In images of mosaically spaced cells, the domains are relatively uniform in size, while images of more irregularly spaced cells display a greater variance of domain areas. Calculating the ratio of the variance to the mean of these areas controls for overall cell density (*Khiripet et al., 2012*). Finally, nearest neighbor analysis measures the distance from each cell to its nearest neighbor. The nearest neighbor regularity index (NNRI) is calculated for each image by dividing the mean nearest neighbor distance by the standard deviation. To control for cell density, the measured NNRI is divided by the NNRI from a randomly generated array of an equal number of points (*Keeley and Reese, 2014*; *Rodieck, 1991*). p-values from all pairwise comparisons are in *Supplementary file 1*.

Unexpectedly, when these tests were applied to DA and bNOS-positive cells, we found a differential dependence on the PDZ-binding domain. DA cells had a significant increase in cell density and a loss of mosaic spacing in *Dscam*^ΔC/ΔC^ mice (*Figure 2D–G*, *Figure 2—figure supplement 1*). By all measures, DA cell organization in *Dscam*^ΔC/ΔC^ retinas was indistinguishable from that of *Dscam*^-/-^ mice. Conversely, bNOS-positive amacrine cells in *Dscam*^ΔC/ΔC^ mice showed little, if any, disruption in organization. bNOS-positive amacrine cells maintained a clear exclusion zone in

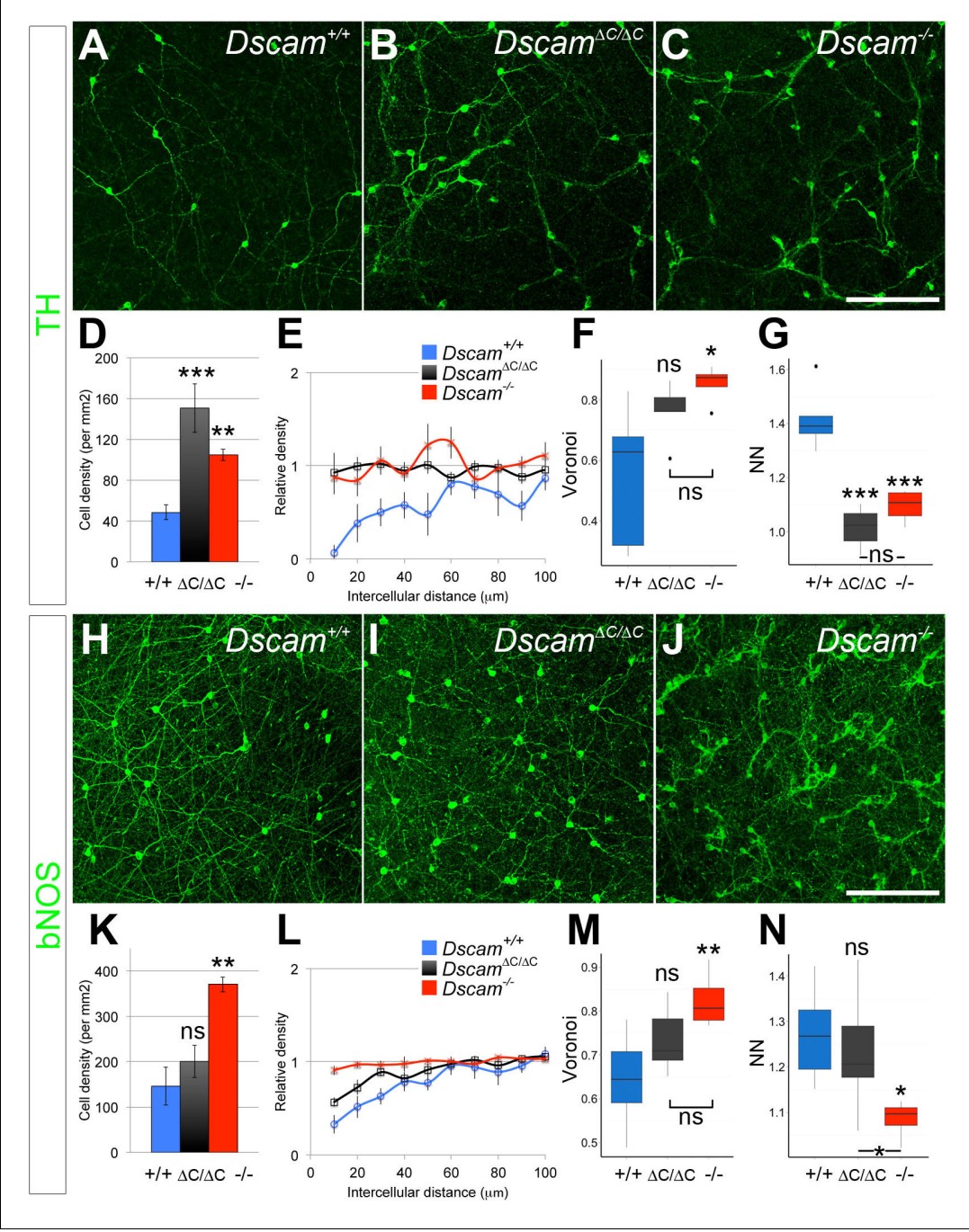

**Figure 2.** DSCAM-mediated self-avoidance requires C-terminal interactions in only some amacrine cell types. (A–C) Dopaminergic amacrine cells (stained for tyrosine hydroxylase, TH) are non-randomly spaced in wild type retinas (A), but lose mosaic spacing and form neurite fascicles in two-week old $Dscam^{\Delta C/\Delta C}$ (B) and $Dscam^{-/-}$ (C) retinas. (D) In both mutants, there was a significant increase in cell density. Spacing was quantified by DRP analysis; relative cell densities normalized to the overall density at increasing distances from reference cells are plotted in (E). By Voronoi (F) and nearest neighbor (G) analyses spacing in $Dscam^{\Delta C/\Delta C}$ retinas was not significantly different than in $Dscam^{-/-}$ animals. (H–J) Conversely, bNOS-positive amacrine cells were not visibly different between controls (H) and $Dscam^{\Delta C/\Delta C}$ retinas (I) despite clear fasciculation and loss of mosaic spacing in $Dscam^{-/-}$ mice (J). $Dscam^{\Delta C/\Delta C}$ values were intermediate between control and $Dscam^{\Delta C/\Delta C}$ in cell density (K), DRP (L), Voronoi (M), and nearest neighbor (N) analyses, but differences from control were not statistically significant. Means ± s.e.m. are represented in D–E, K–L. Box plots represent the median, first and third quartile, range, and outliers. N = 4–8 retinas per genotype. *p<0.05; **p<0.01; ***p<0.001; n.s. is not significant by Tukey post-hoc

*Figure 2 continued on next page*

*Figure 2 continued*

test between indicated genotypes or compared to controls. Scale bar is 100 µm. Representative Voronoi domains are in *Figure 2—figure supplement 1*.

The following figure supplement is available for figure 2:

**Figure supplement 1.** Examples of Voronoi tessellation domains in *Dscam* mutants.

$Dscam^{\Delta C/\Delta C}$ mice, and were not significantly different from wildtype controls by any spacing measure, nor did their overall cell density differ from controls (*Figure 2K–N*, *Figure 2—figure supplement 1*).

To ask if this differential dependence on the PDZ-interacting C-terminus extended to other cell types, we analyzed two populations of RGCs: intrinsically photosensitive retinal ganglion cells (ipRGCs), and cells labeled in the *Cdh3-GFP* GENSAT transgenic line (*Osterhout et al., 2011*). ipRGCs include as many as five distinct populations which can be differentiated by morphology, but only three of these subtypes – M1, M2, and M3 – are labeled by an antibody to melanopsin postnatally (*Schmidt et al., 2011*). M1 cells stratify their dendrites to the outermost lamina in the OFF region of the inner plexiform retina and stain the brightest for melanopsin. M2 cells stratify in the ON layer much closer to the RGC cell bodies. M3 cells are bistratified, but are much more rare than the M1 or M2 population. ipRGCs are among the most clustered and fasciculated cell types in $Dscam^{-/-}$ mutants; both M1 and M2 cells are severely clustered with tight dendritic fascicles (*Figure 3A,C*, and [*Fuerst et al., 2009*]). In $Dscam^{\Delta C/\Delta C}$ mutants, cell bodies were clearly clustered, albeit not as severely as in $Dscam^{-/-}$ mice (*Figure 3B,E–G*, *Figure 3—figure supplement 1*). Cell density appeared to be modestly increased compared to controls, but the difference was not statistically significant (*Figure 3D*). Thus, in general, ipRGCs were intermediately affected. The Cdh3-GFP RGC cell number was increased in $Dscam^{-/-}$ mutants, and the cell bodies aggregated into clusters (*Figure 3H,J*). In contrast, cell number was not significantly increased in $Dscam^{\Delta C/\Delta C}$ retinas, and the cell body clustering observed in $Dscam^{-/-}$ mutants was not observed (*Figure 3I,K–N*, *Figure 3—figure supplement 1*). Thus, in both amacrine and ganglion cells, some cell types require the PDZ-interacting C-terminus of DSCAM for its function, whereas in other cell types, this domain is largely dispensable. Furthermore, the functions of DSCAM in promoting cell death and self-avoidance change in parallel and are not separated by the perturbation of this intracellular interaction.

One possible explanation for this differential dependence could be that DSCAM-ΔC protein is fully functional, but selectively unstable in some cell types, such as DA cells and ipRGCs, which are affected by the C-terminal deletion, but maintained at normal levels in other cell types, such as bNOS-positive amacrine cells, which are largely unaffected by the C-terminal deletion. We used two methods to assess selective instability. First, we quantified the colocalization between DSCAM and melanopsin to see if there was a reduction in DSCAM specifically in ipRGCs. We did not find any difference in this colocalization between $Dscam^{+/+}$ and $Dscam^{\Delta C/\Delta C}$ retinas (*Figure 1—figure supplement 2*). Second, we quantified the fluorescence intensity of *Dscam* staining in cryosections along a 10 µm line perpendicular to the INL directly adjacent to this cellular layer. This region includes the S1 lamina in which DA cells stratify their neurites. No reduced fluorescence intensity was detectable in $Dscam^{\Delta C/\Delta C}$ retinas, whereas we could detect a trend towards reduced labeling in heterozygous $Dscam^{+/-}$ images relative to $Dscam^{+/+}$ controls, (*Figure 1—figure supplement 1E–I*). Thus, neither immunofluorescence nor immunoprecipitation (*Figure 1B*) provide any evidence supporting protein instability. It remains possible, however, that a subtle difference in cell-type specific protein stability or localization beyond the resolution of these experiments could contribute to the range of phenotypes we have observed. Similarly, the tag could be interfering with function in some cell types; however, we do not believe this to be the case as we performed similar experiments with DSCAML1 using a different tag and obtained similar results, as described below.

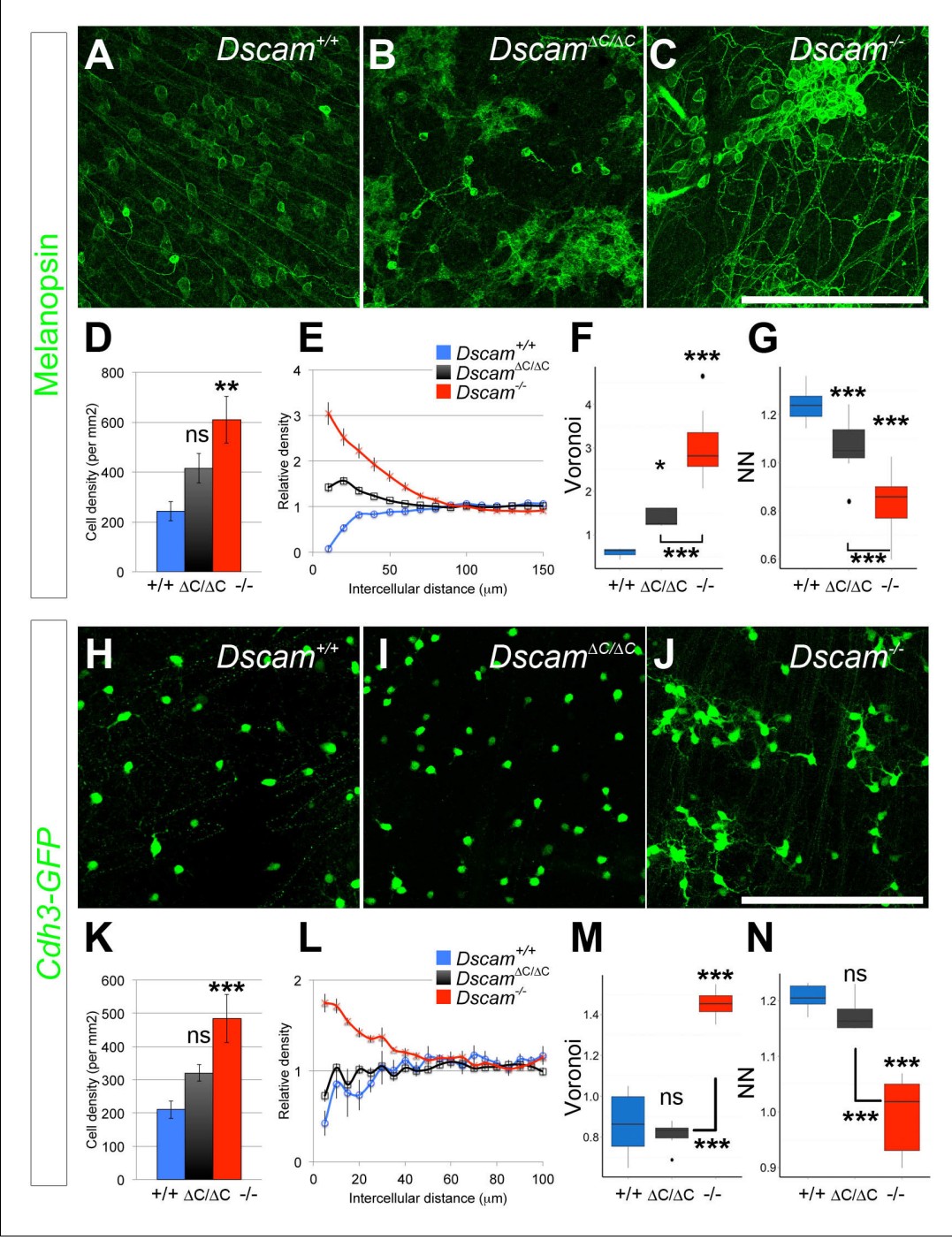

**Figure 3.** RGCs also display differential dependence on DSCAM C-terminal interactions for self-avoidance. (**A–C**) Melanopsin-positive intrinsically photoresponsive retinal ganglion cells are found in a mosaic pattern in wild type retinas (**A**), but at two weeks of age, ipRGC cell bodies in *Dscam*^ΔC/ΔC^ (**B**) retinas are pulled into clusters similar to those in *Dscam*^-/-^ (**C**) retinas. (**D**) Overall ipRGC density was not significantly increased in *Dscam*^ΔC/ΔC^ retinas. (**E**) By DRP, cell body clustering was intermediate between *Dscam*^+/+^ and *Dscam*^-/-^ retinas. Voronoi (**F**) and nearest neighbor (**G**) analyses also revealed a clear intermediate defect. Values from *Dscam*^ΔC/ΔC^ retinas were significantly different from both control and *Dscam*^-/-^ mutants. (**H–J**) Cdh3-GFP RGCs are mosaically spaced in control retinas (**H**), and this spacing is not perturbed in the *Dscam*^ΔC/ΔC^ retinas (**I**), but these cells form clusters in *Dscam*^-/-^ animals (**J**), indicating interactions mediated by DSCAM's C-terminus are dispensable to prevent these cells from clustering. (**K**) Cdh3-GFP RGC overall cell density was significantly increased in *Dscam*^-/-^ retinas, but not in *Dscam*^ΔC/ΔC^ mutants. (**L**) A clear exclusion zone is detectable by DRP analysis in *Dscam*^+/+^ and *Dscam*^ΔC/ΔC^ retinas. *Figure 3 continued on next page*

*Figure 3 continued*
This exclusion zone is lost in *Dscam*<sup>-/-</sup> animals, where cell density is increased adjacent to any given cell, indicative of clustering. Similarly, Voronoi (**M**) and nearest neighbor (**N**) analyses describe a clear spacing defect in *Dscam*<sup>-/-</sup> but not in *Dscam*<sup>ΔC/ΔC</sup> retinas. Means ± s.e.m. are represented in D–E, K–L. Box plots represent the median, first and third quartile, range, and outliers. N = 6 retinas per genotype. *p<0.05; **p<0.01; ***p<0.001; n.s. is not significant by Tukey post-hoc test between indicated genotypes or compared to controls. Scale bar is 250 μm. Representative Voronoi domains are in *Figure 3—figure supplement 1*.
The following figure supplement is available for figure 3:

**Figure supplement 1.** Examples of Voronoi tessellation domains in *Dscam* mutants.

## Dependence on DSCAML1 C-terminal interactions also varies with cell type

*Dscam* and *Dscaml1* serve similar functions in the retina, but in different cell types based on their non-overlapping expression patterns. *Dscaml1* is expressed in the cells that contribute to the rod circuit, including rod photoreceptors, rod bipolar cells (RBCs), and AII amacrine cells (*Fuerst et al., 2009*), as well as a population of previously undefined cells in the inner nuclear layer. To test if DSCAML1 functions, like DSCAM, require C-terminus-mediated PDZ-interactions in only some cell types, we created mice harboring a similar replacement of the final ten amino acids of DSCAML1 with an epitope tag (HA) using the same strategy as described for DSCAM (*Figure 4A*). DSCAML1-Δ C protein was stable and localized to the cell surface in transfected HEK293T cells. In a surface biotinylation assay, comparable proportions of DSCAML1-ΔC and full-length DSCAML1 proteins were at the surface (not shown). DSCAML1-ΔC protein had the expected membrane topology, but did not co-immunoprecipitate with MAGI-3 (*Figure 4—figure supplement 1*). Like *Dscam*, *Dscaml1*<sup>ΔC/ΔC</sup> retinas were modestly expanded, but not so severely as in *Dscaml1*<sup>-/-</sup> mutants (*Figure 4B–D*).

We assessed the density and spacing of two cell types: AII amacrine cells and VGLUT3-positive amacrine cells. Both cell types increase in number and lose mosaic spacing in *Dscaml1*<sup>-/-</sup> mutants (*Figure 4E,G,L,N* and [*Fuerst et al., 2009*]). AII amacrine cells in *Dscaml1*<sup>ΔC/ΔC</sup> mice were subtly disrupted, but more similar to the control condition than to that of *Dscaml1*<sup>-/-</sup> animals (*Figure 4F*). There was no increase in cell density, and cells did not form clusters in *Dscaml1*<sup>ΔC/ΔC</sup> mice, as they did in *Dscaml1*<sup>-/-</sup> animals, as measured by DRP or nearest neighbor analysis (*Figure 4H,I,K*). There was, however, an increased covariance of Voronoi tessellation domain areas compared to controls (*Figure 4J*, *Figure 4—figure supplement 2*) indicative of irregular spacing. Conversely, VGLUT3-positive amacrine cells in *Dscaml1*<sup>ΔC/ΔC</sup> mice were more similar to *Dscaml1*<sup>-/-</sup> than to controls (*Figure 4L–N*). Cell number was significantly increased, and DRP, Voronoi, and nearest neighbor analyses all indicated a significant loss of mosaic spacing (*Figure 4O–R*, *Figure 4—figure supplement 2*). Thus, as with DSCAM, some cell types have a greater dependence on DSCAML1 C-terminus than do others.

## Increased cell number is not sufficient to explain self-avoidance defects in ΔC mutants

Increased cell number from a lack of cell death can disrupt mosaic spacing in some cell types, although not as severely as *Dscam*<sup>-/-</sup> mutation (*Keeley et al., 2012*). To ask how much the increased cell number in ΔC mutants contributed to abnormal spacing, we compared the cell types affected in *Dscam*<sup>ΔC/ΔC</sup> (ipRGCs and DA cells) and *Dscaml1*<sup>ΔC/ΔC</sup> (VGLUT3-positive amacrine cells) to *Bax*<sup>-/-</sup> mutants. In all three cases, cell density was, if anything, higher in *Bax*<sup>-/-</sup> retinas than in ΔC mutants (*Figure 5A,E,I*), although this was only significant for VGLUT3 cells. ipRGCs were more severely clustered in *Dscam*<sup>ΔC/ΔC</sup> than in *Bax*<sup>-/-</sup> animals, as measured by DRP packing factor (*Figure 5B*) and Voronoi domain analysis (*Figure 5C*), but were not significantly different by nearest neighbor analysis (*Figure 5D*). DA cell spacing was similarly disrupted in *Bax*<sup>-/-</sup>mutants as in *Dscam*<sup>ΔC/ΔC</sup> (*Figure 5F–H*), while VGLUT3-positive cells in *Dscaml1*<sup>ΔC/ΔC</sup> were significantly more disrupted by Voronoi and nearest neighbor analyses, but not by DRP packing factor, despite a significantly lower cell density than *Bax*<sup>-/-</sup> retinas (*Figure 5I–L*). Therefore, cell spacing of ipRGCs and

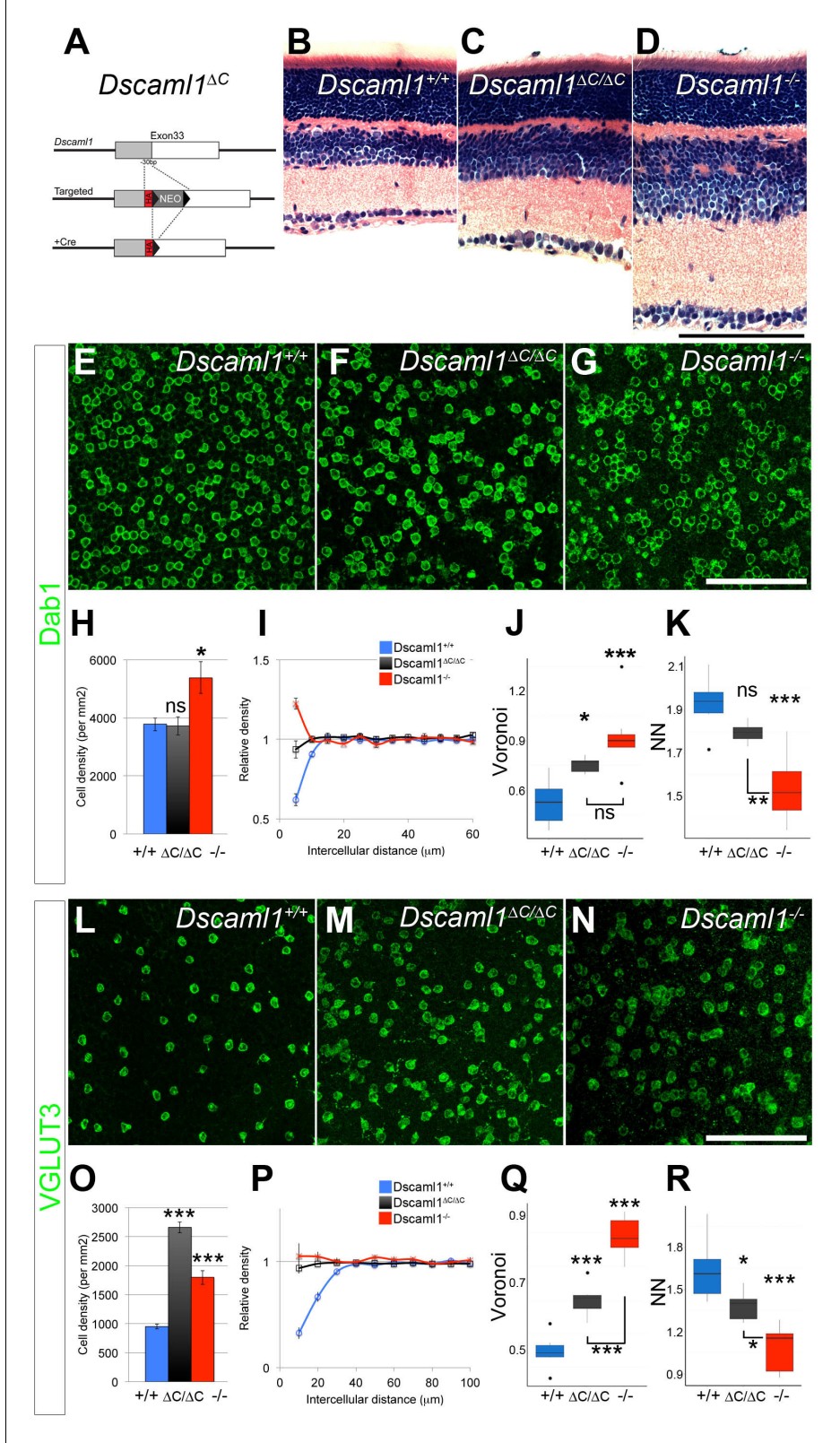

**Figure 4.** DSCAML1-mediated self-avoidance requires C-terminal interactions in only some cell types. (**A**) *Dscaml1^(ΔC/ΔC)^*mutant mice were generated by replacing the sequence encoding the final ten amino acids with an HA tag by homologous recombination. See also *Figure 4—figure supplement 1*. (**B–D**) Hematoxylin and eosin staining of adult retinas shows that, compared to controls (**B**), *Dscaml1^-/-^* retinas (**D**) are significantly expanded and

*Figure 4 continued on next page*

*Figure 4 continued*

disorganized. $Dscaml1^{\Delta C/\Delta C}$ retinas (C) have a more intermediate expansion without the extensive disorganization found in the null mutant. (E–G) AII amacrine cells (Dab1-positive) are organized in a mosaic pattern in two-week old control retinas (E). This pattern is disrupted in $Dscaml1^{\Delta C/\Delta C}$ retinas (F), but not as severely as in $Dscaml1^{-/-}$ retinas (G), where the cells form clusters. (H) There was a significant increase in cell density in $Dscaml1^{-/-}$ but not in $Dscaml1^{\Delta C/\Delta C}$ animals. I) DRP analysis revealed an intermediate effect in $Dscaml1^{\Delta C/\Delta C}$ retinas between the clear exclusion zone in control and clustering in $Dscaml1^{-/-}$. AII amacrine spacing was slightly disrupted in $Dscaml1^{\Delta C/\Delta C}$ retinas by Voronoi analysis (J) but not by nearest neighbor analysis (K). L-N) Conversely, compared to controls (L) the disruption of VGLUT3-positive amacrine cell spacing in $Dscaml1^{\Delta C/\Delta C}$ (M) retinas was more similar to that in $Dscaml1^{-/-}$ (N) retinas. (O) VGLUT3-positive amacrine cell density was significantly increased in both $Dscaml1^{\Delta C/\Delta C}$ and $Dscaml1^{-/-}$ animals. (P) DRP analysis reveals the loss of exclusion zone in both mutants. (Q,R) $Dscaml1^{\Delta C/\Delta C}$ values were significantly different from controls in both Voronoi and nearest neighbor analyses. Means ± s.e.m. are represented in H–I, O–P. Box plots represent the median, first and third quartile, range, and outliers. N = 6–8 retinas per genotype. *p<0.05; ***p<0.001; n.s. is not significant by Tukey post-hoc test between indicated genotypes or compared to controls. Scale bars are 100 μm. Representative Voronoi domains are in *Figure 4— figure supplement 2*.

The following figure supplements are available for figure 4:

**Figure supplement 1.** DSCAML1-ΔC has a normal membrane topology, but does not interact with MAGI-3.

**Figure supplement 2.** Examples of Voronoi tessellation domains in *Dscaml1* mutants.

---

VGLUT3-positive amacrine cells was more severely disrupted in ΔC mutants than could be explained by increased cell number alone.

The exception to this observation was DA cells, in which case the increased density in $Bax^{-/-}$ retinas disrupted cell body mosaics to a similar extent as loss of the C-terminus of DSCAM (*Figure 5F–H*). However, we observed that the DA neurites were not severely fasciculated in $Bax^{-/-}$ mutants (*Figure 6D*). To separately evaluate neurite fasciculation, we developed a method to quantify fasciculation. In this technique, observers blind to genotype were asked to choose which of two randomly presented images was less fasciculated without regard to cell number. A score was generated for each image based on iterative head-to-head matchups by an Elo algorithm (*Elo, 1978*), which uses these win-loss matchups to efficiently sort the images (Elo score, see Materials and methods). Images that the observers consistently deemed less fasciculated received high scores, while those deemed more fasciculated received low scores. The mean score per retina was used to compare across genotypes by a Wilcoxon rank sum test with the Benjamini and Hochberg correction. For DA cells, there was significantly more severe fasciculation in $Dscam^{\Delta C/\Delta C}$ and $Dscam^{-/-}$ retinas than in controls or $Bax^{-/-}$ retinas (*Figure 6A–D,M*).

We next applied this technique for grading dendrite fasciculation to ipRGCs by collapsing individual z-stacks of images into distinct projections through the ON portion (*Figure 6E–H*) and through the OFF portion (*Figure 6I–L*) of the IPL. Here, clear differences between $Dscam^{\Delta C/\Delta C}$ and $Dscam^{-/-}$ retinas were visible. In the ON layer (predominantly M2 cells) dendritic labeling was denser directly above clustered cell bodies, but spread out to evenly cover the field in $Dscam^{\Delta C/\Delta C}$ images (*Figure 6F*). The Elo score could not distinguish $Dscam^{\Delta C/\Delta C}$ retinas from control or from $Bax^{-/-}$ retinas, but $Dscam^{-/-}$ fasciculation was significantly more severe than in all other genotypes (*Figure 6G, N*). In the OFF layer (predominantly M1 cells) there were smaller and fewer fascicles in $Dscam^{\Delta C/\Delta C}$ than in null retinas (*Figure 6J–K*). Indeed, by Elo score $Dscam^{\Delta C/\Delta C}$ retinas were intermediately fasciculated between control and $Dscam^{-/-}$ but were indistinguishable from $Bax^{-/-}$ mutants (*Figure 6O*).

We also assessed the fasciculation of Cdh3-GFP RGC dendrites and found tight fasciculation in $Dscam^{-/-}$ mutants, but no discernable difference between $Dscam^{\Delta C/\Delta C}$ and control retinas (*Figure 6— figure supplement 1*). Similarly, bNOS-positive amacrine cells had mild fasciculation, if any, in $Dscam^{\Delta C/\Delta C}$ and $Bax^{-/-}$ mutants (*Figure 6—figure supplement 1*).

Thus, increased cell density can contribute to abnormal spacing and dendrite fasciculation in some cell types as previously reported (*Keeley et al., 2012*), but is not sufficient to explain the self-avoidance deficits and especially the fasciculation of processes found $Dscam^{-/-}$ retinas, and to varying degrees, in ΔC mutant retinas.

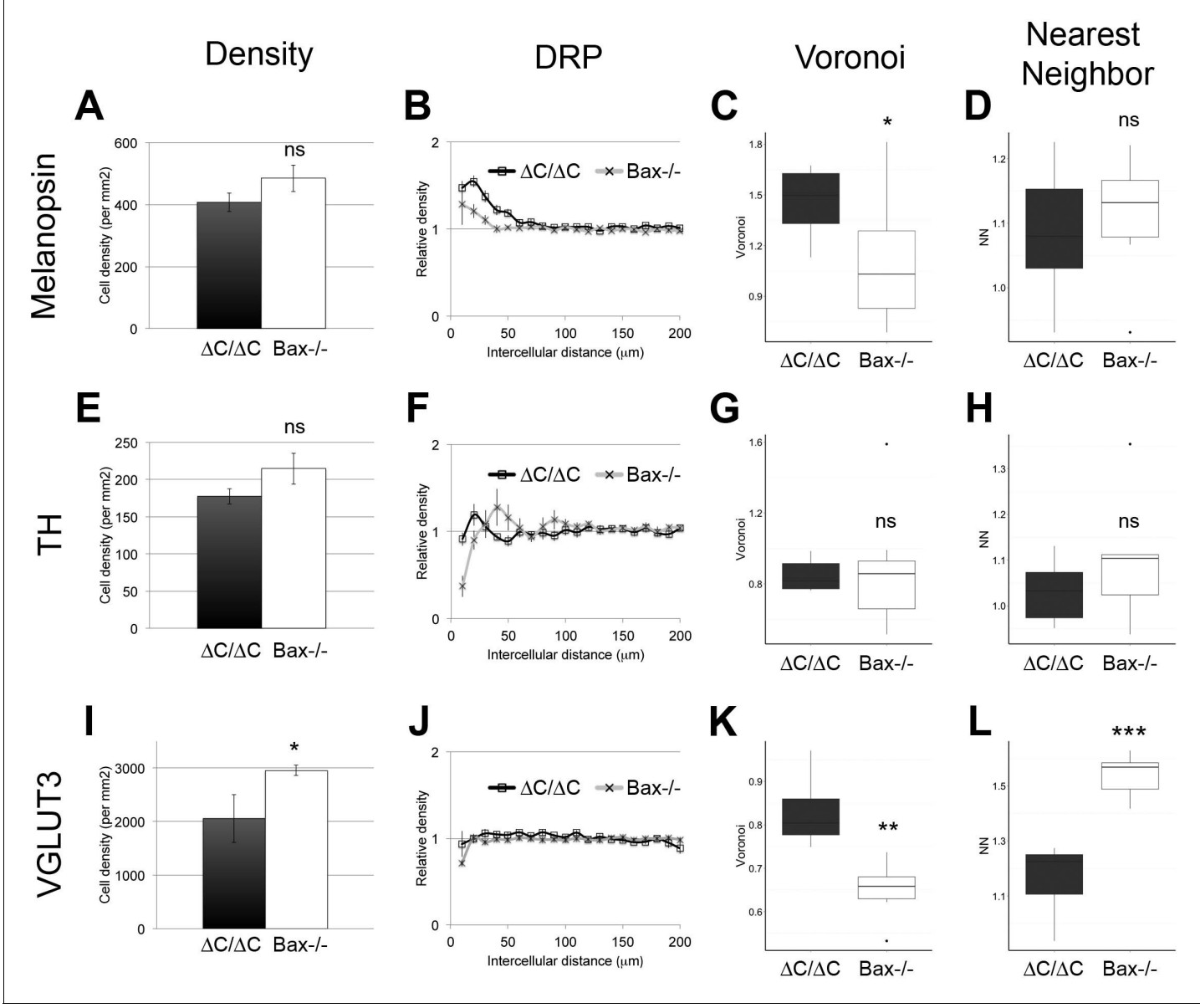

**Figure 5.** Increased cell density is not sufficient to explain spacing defects. (**A**) ipRGC cell density was similar in *Dscam*^ΔC/ΔC^ and *Bax*^-/-^ retinas. Despite this, clustering was more severe in *Dscam*^ΔC/ΔC^ as measured by DRP (**B**) and Voronoi (**C**), but not by nearest neighbor analysis (**D**). (**E**) Likewise, DA cell density was similar in *Dscam*^ΔC/ΔC^ and *Bax*^-/-^ retinas. However, DA cell spacing was not significantly different between *Dscam*^ΔC/ΔC^ and *Bax*^-/-^ mutants (**F–H**). (**I**) VGLUT3-positive amacrine cell density was significantly higher in *Bax*^-/-^ than in *Dscaml1*^ΔC/ΔC^ retinas. Mosaic spacing was more disrupted in *Dscaml1*^ΔC/ΔC^ as measured by Voronoi domain analysis (**K**) and nearest neighbor (**L**) but not by DRP (**J**). Means ± s.e.m. are represented in **A–B**, **E–F**, **I–J**. Box plots represent the median, first and third quartile, range, and outliers. *p<0.05; **p<0.01; ***p<0.001; n.s. is not significant by student's t-test.

## PDZ-interacting C-termini promote neurite stratification in some cell types

Dscams promote normal neurite stratification in some cell types, although this is sensitive to genetic background. bNOS-positive amacrine cells are largely disorganized in *Dscam* mutants on a C3H background, but not in mixed C57BL/6 – BALBc strains (*Fuerst et al., 2010*). This has not been assessed in null mutants on a clean C57BL/6 background, as these mice die at birth (*Amano et al., 2009*). We inspected bNOS-positive amacrine stratification in *Dscam*^ΔC/ΔC^ retinas, which are on a C57BL/6 background, and found no abnormalities (*Figure 7A–B*). DA cell and M1 ipRGCs co-stratify

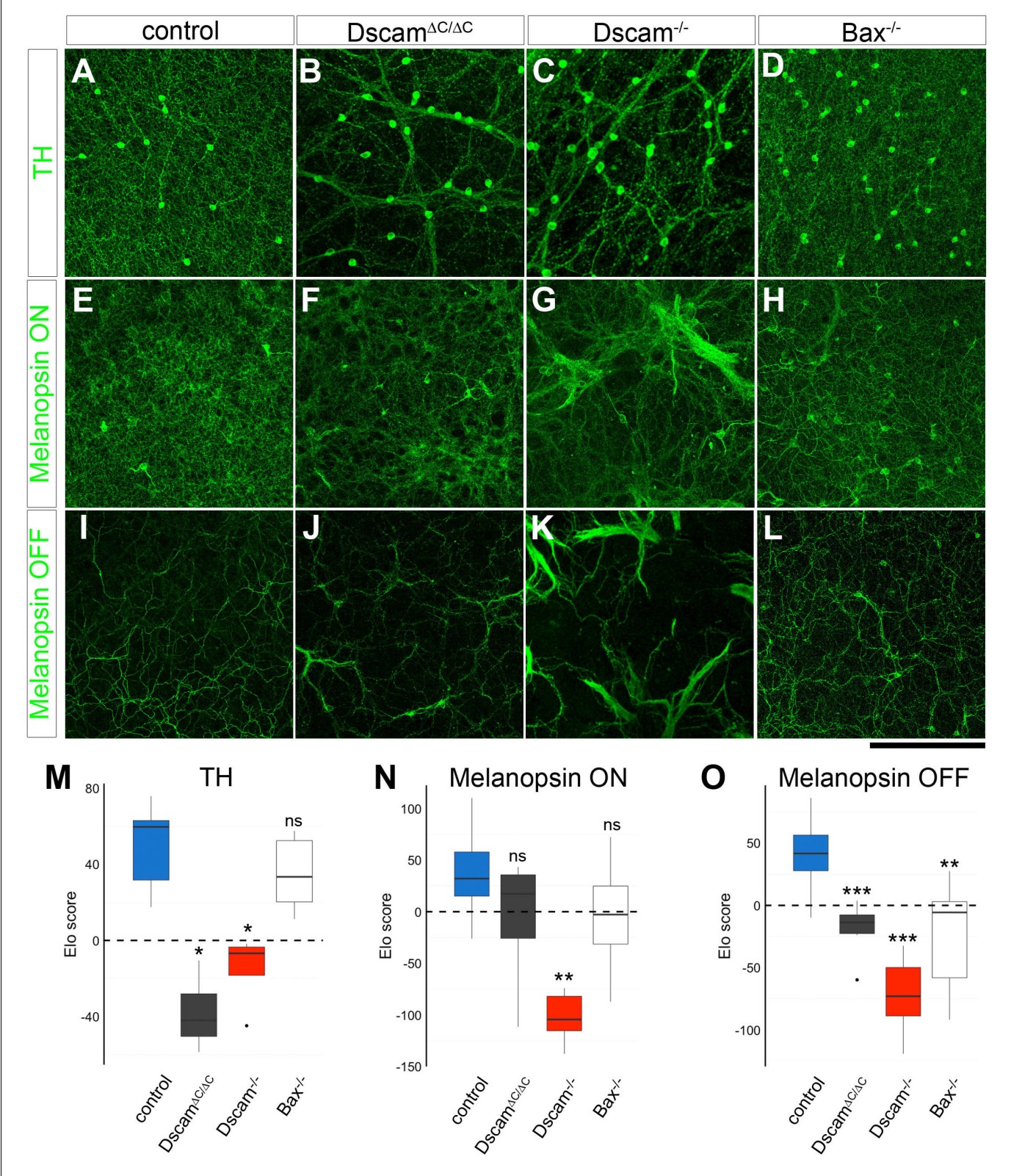

**Figure 6.** Neurite fasciculation is separable from density-dependent cell body clustering. (A) DA cell neurites evenly fill their receptive fields in control mice, but form fascicles in *Dscam*$^{\Delta C/\Delta C}$ (B) and *Dscam*$^{-/-}$ (C) animals. (D) DA fascicles are rarely observed in *Bax*$^{-/-}$ mutants. (E) M2 ipRGC dendrites imaged in the ON region of the IPL are evenly distributed in wild type mice and largely remain so in *Dscam*$^{\Delta C/\Delta C}$ (F) and *Bax*$^{-/-}$ (H) mutants, while severe fasciculation is evident in *Dscam*$^{-/-}$ retinas (G). I) In the OFF strata of the IPL, M1 ipRGC dendrites are diffusely organized. There is modest fasciculation

*Figure 6 continued on next page*

*Figure 6 continued*

in *Dscam*$^{\Delta C/\Delta C}$ (J) and *Bax*$^{-/-}$ (L) mice, while fasciculation in *Dscam*$^{-/-}$ retinas (K) is much more severe. (M–O) Elo ranking of fasciculation severity between genotypes demonstrates that DA neurites (M) are clearly fasciculated in *Dscam*$^{\Delta C/\Delta C}$ and *Dscam*$^{-/-}$ retinas, but not in *Bax*$^{-/-}$, which had loss of mosaic cell body spacing. Conversely, ipRGCs in *Dscam*$^{\Delta C/\Delta C}$ did not have significantly more fasciculation than *Bax*$^{-/-}$ either in the ON (N) or OFF (O) layers, despite having a more severe cell body clustering. Box plots represent the median, first and third quartile, range, and outliers. *p<0.05; **p<0.01; ***p<0.001; n.s. is not significant by Wilcoxon rank sum test. Scale bar is 250 μm. See also *Figure 6—figure supplement 1*.

The following figure supplement is available for figure 6:

**Figure supplement 1.** Fasciculation of Cdh3-GFP RGCs and bNOS-positive amacrine cells.

correctly in the OFF region of the IPL in *Dscam*$^{-/-}$ mutants (*Fuerst et al., 2009*), and both correctly targeted in *Dscam*$^{\Delta C/\Delta C}$ mutants as well (*Figure 7C–D*). DA and M1 ipRGCs provide the only clear example we have seen of co-fasciculation in *Dscam*$^{-/-}$ mutants (*Fuerst et al., 2009*). In *Dscam*$^{\Delta C/\Delta C}$ mutants, DA cells are severely fasciculated (*Figure 6M*), while M1 ipRGCs more mildly so (*Figure 6O*). Interestingly, there is a clear co-fasciculation between these cell types in *Dscam*$^{\Delta C/\Delta C}$ mutants (*Figure 7E*), suggesting that the M1 fasciculation may be influenced by the DA cells.

In *Dscaml1*$^{-/-}$ mutants, stratification and synaptic pairing between rod bipolar cells and AII amacrine cells is preserved (*Fuerst et al., 2009*). We found that to be the case in *Dscaml1*$^{\Delta C/\Delta C}$ mice as well: both cell types project to the ON region of the IPL adjacent to the RGL (*Figure 7F–I*) where they connect at dyad synapses. In *Dscaml1*$^{-/-}$ mice, these synapses displayed features reminiscent of immature synapses, including malformed ribbons and an overabundance of synaptic vesicles (*Figure 7K* and [*Fuerst et al., 2009*]). We inspected the dyad synapses in *Dscaml1*$^{\Delta C/\Delta C}$ retinas by transmission electron microscopy, and failed to detect these synaptic phenotype reminiscent of the *Dscaml1*$^{-/-}$ mice (*Figure 7L–M*), indicating that DSCAML1 promotes synapse maturation independent of C-terminal interactions.

We determined VGLUT3-positive amacrine cells depend on DSCAML1 C-terminal interactions for self-avoidance and normal developmental cell death (*Figure 4*). This population stratifies its neurites to the ON-OFF region of the IPL between the ChAT-positive laminae in wild type mice (*Figure 7N*). We found that in *Dscaml1*$^{\Delta C/\Delta C}$ mutants, some of these neurites projected past the ON ChAT band into the region where AII amacrines and RBCs target (*Figure 7O*). Indeed, much of the VGLUT3 labeling below the ON ChAT band was directly adjacent to Dab1-positive AII amacrine terminals (*Figure 7P*), an association that persisted through adulthood (18 months, *Figure 7Q*). We quantified this misprojection in whole mount retinas stained for VGLUT3 and Dab1 imaged *en face*. We made projections through the Dab1-positive terminals blind to the VGLUT3 channel and to genotype, then quantified the area occupied by VGLUT3 labeling at three threshold levels. VGLUT3 occupied a significantly greater percentage of area in *Dscaml1*$^{\Delta C/\Delta C}$ animals than in controls (2-way ANOVA, p<0.0001, *Figure 7R*).

Thus, in ΔC mutants, we found cell types displaying defects in all three categories of Dscam function – cell death, self-avoidance, and neurite stratification – as well as cell types with relatively little dysfunction.

## Discussion

We tested the functional significance of the Dscams' PDZ-interacting domains in vivo by replacing the C-terminal ten amino acids with epitope tags (*Dscam*$^{\Delta C}$ and *Dscaml1*$^{\Delta C}$). The Dscams have similar functions in the different cell types in which they are expressed: (1) They promote developmental cell death, (2) they promote self-avoidance at the cell type level both between cell bodies and neurites, preventing cell body clustering and fasciculation of processes, and (3) they can promote the laminar specificity of neurite stratification and synapse maturation (*Fuerst et al., 2009, 2008*). We hypothesized that interactions with PDZ domain-containing proteins via the C-termini would mediate a subset of these functions. Instead, we found that a subset of cell types required the PDZ interaction for all of these functions; while in other cell types these processes could proceed relatively normally without the Dscams' C-termini. The phenotypes found in ΔC mutants compared to null mutants are summarized by cell type in *Table 1*.

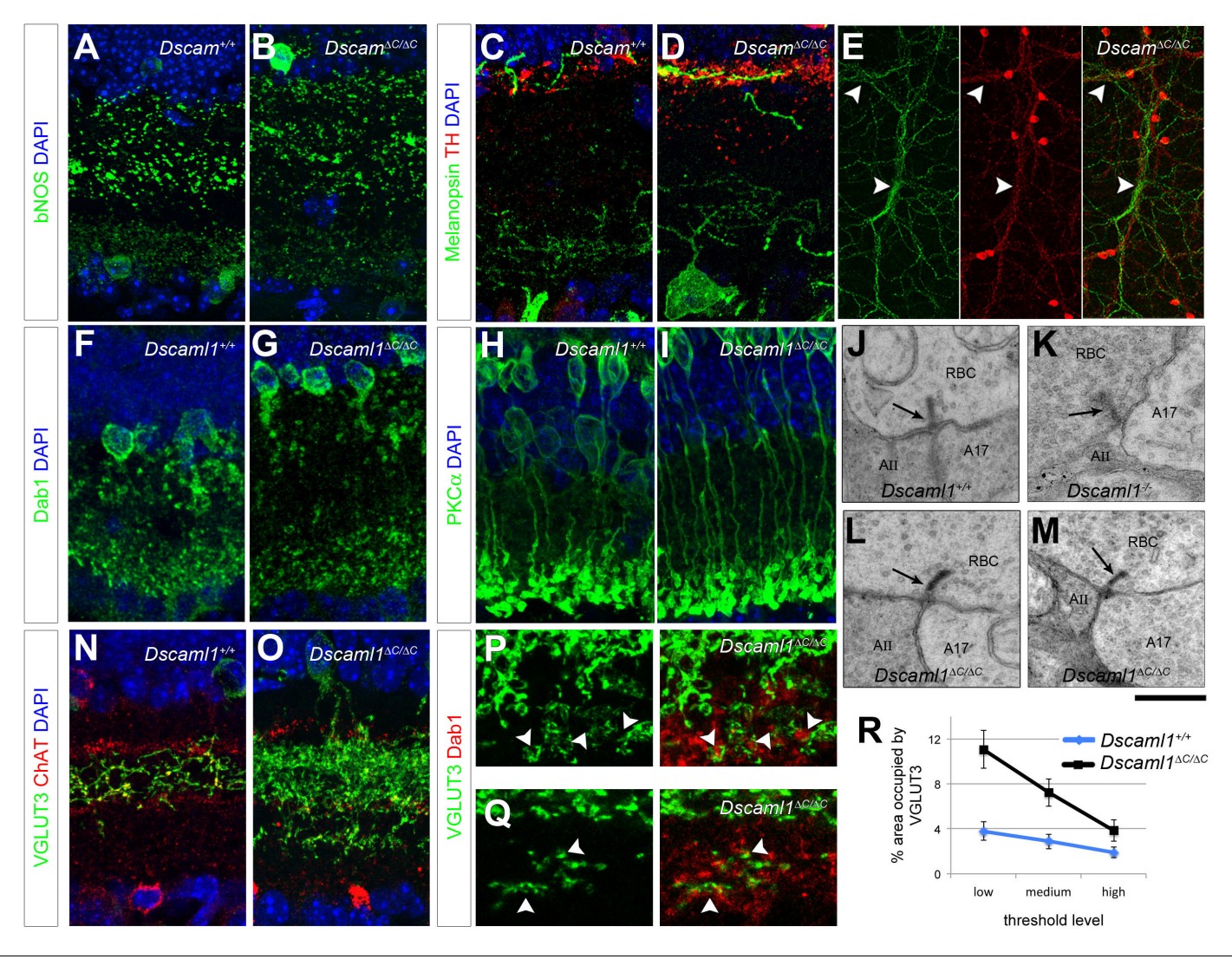

**Figure 7.** Laminar specificity in ΔC mutants. Neurite stratification in the IPL was analyzed in immunolabeled cryosections. (A,B) bNOS-positive amacrine cells stratified normally in $Dscam^{\Delta C/\Delta C}$ retinas, as did ipRGCs and DA cells (C,D), which co-stratify in the OFF region adjacent to the INL. (E) Imaged *en face*, DA neurites co-fasciculated with ipRGC dendrites in $Dscam^{\Delta C/\Delta C}$ mutants (arrowheads), as we have previously found in $Dscam^{-/-}$ retinas (**Fuerst et al., 2009**). In $Dscaml1^{\Delta C/\Delta C}$ mutants, AII amacrine cells (F,G) and rod bipolar cells (H,I) terminate their processes normally. (J–M) TEM analysis revealed that $Dscaml1^{\Delta C/\Delta C}$ retinas contained structurally normal dyad synapses between rod bipolar cells and AII/A17 amacrine cells with distinct ribbons (arrows). (K) $Dscaml1^{-/-}$ RBC dyad synapses are characterized by excess in synaptic vesicle number and indistinct ribbons. Four retinas analyzed by TEM per genotype, > 10 synapses inspected per retina. (N,O) VGLUT3-positive amacrine cells misprojected beyond the ON ChAT layer in $Dscaml1^{\Delta C/\Delta C}$ mutants. (P,Q) These ectopic neurites became associated with AII amacrine terminals adjacent to the retinal ganglion cell layer (arrowheads). This association was observable at 3 weeks of age (P) and persisted through adulthood (Q, 18 months of age). (R) These misprojections were quantified by imaging whole-mount retinas stained for VGLUT3 and Dab1 *en face* and calculating the percent of area occupied by VGLUT3 in projections through Dab1-positive AII amacrine terminals. Means ± s.e.m. at three threshold levels are represented in R. n = 6–8 retinas per genotype. Scale bar is 20 µm in A–O, 110 µm in E, 10 µm in P,Q, and 500 nm in J–M.

Phrased genetically, the $Dscam^{-/-}$ and $Dscaml1^{-/-}$ alleles represent a complete loss of function. The functions we have defined, promoting developmental cell death, promoting self-avoidance in both processes and cell bodies to enable mosaic spacing and coverage factor, and promoting laminar specificity and synapse maturation in at least some cell types and genetic backgrounds, are based on the loss of function mutant phenotype. The deletion of the PDZ-binding C-termini of DSCAM and DSCAML1 had several possible outcomes in this context. It could have resulted in an

**Table 1.** Phenotypes by cell type in $\Delta C$ mutants. The phenotypes assessed in $Dscam^{\Delta C/\Delta C}$ and $Dscaml1^{\Delta C/\Delta C}$ are summarized. n.a. is not applicable, n.s. is not shown.

| Cell type | Cell spacing | Cell density | Fasciculation | Neurite stratification |
|---|---|---|---|---|
| DA | nearly null | nearly null | nearly null | normal |
| bNOS | nearly wild type | nearly wild type | nearly wild type | normal |
| Cdh3-GFP | nearly wild type | nearly wild type | nearly wild type | normal (n.s.) |
| ipRGC | Intermediate | nearly wild type | Intermediate | normal |
| VGLUT3 | nearly null | nearly null | n.a. | misprojection |
| AII | Intermediate | nearly wild type | n.a. | normal |

unstable or mis-trafficked protein, causing a complete loss of function. Our data do not support this, as protein is found at normal levels by both immunoprecipitation and immunohistochemistry, is on the cell surface by live cell staining and surface biotinylation, and is in the correct topological orientation, although more subtle or specific defects may still have been missed by these analyses. The PDZ-interacting C-termini of DSCAM and DSCAML1 could also have been totally superfluous for its function. Consistent with previous results (*Yamagata and Sanes, 2010*), this also is not the case. A partial loss of function could have resulted in a subset of the phenotypically defined functions being present, and others being lost. Given the interaction with synaptic scaffolding proteins, a phenotype of impaired synapse maturation with normal developmental cell death and self-avoidance was a reasonable anticipated outcome. Interestingly, we instead found a partial loss of function at the level of cell types. Some cell types appear to almost completely require the C-termini, whereas in other cell types they are almost completely dispensable. In yet other cell types, the phenotype was intermediate, between wild type and the null, but all phenotypically defined functions changed roughly in parallel. An additional caveat is that the epitope tags used to replace the PDZ-binding motifs may contribute to these phenotypes. While this is formally possible, we believe that it is unlikely to be the case, as we used two different epitope tags (Myc and HA) with two different genes and found a similar range of phenotypes between different cell types. Thus, our main conclusion from these studies is that the requirement for a PDZ-interacting C-terminus in DSCAM or DSCAML1 is variable and depends on the cell type.

How cell adhesion molecules function to prevent adhesion and promote self-avoidance remains and interesting question. The other well described mediators of self-avoidance function by generating thousands of distinctly homophilic recognition units (*Zipursky and Grueber, 2013*). *Dscam1* in *Drosophila* uses three banks of alternatively spliced exons to produce 19,008 isoforms with distinct extracellular domains (*Schmucker et al., 2000*). Each of these three exons encode regions of the protein essential for homophilic binding, resulting in 19,008 potential molecules that preferentially recognize other copies of the same isoform (*Sawaya et al., 2008*; *Wojtowicz et al., 2004, 2007*; *Wu et al., 2012*). By biased stochastic exon choice, a given neuron expresses an estimated 10–50 isoforms, most of which will differ from neighboring neurons (*Miura et al., 2013*; *Neves et al., 2004*; *Zhan et al., 2004*). This gives each neuron a distinct fingerprint allowing it to recognize and avoid 'self' through repulsion while still interacting with its neighbors, a process called self/non-self discrimination (*Hughes et al., 2007*; *Matthews et al., 2007*; *Soba et al., 2007*). As might be expected with such a mechanism in which each cell is uniquely identified, vast isoform diversity is required for Dscam1 to function normally (*Hattori et al., 2009, 2007*).

Similarly, the vertebrate gamma protocadherin cluster (*Pcdhg*) can also produce diverse recognition units. There are only 22 *Pcdhg* isoforms, specified by alternative promoter choice within the gene cluster instead of alternative splicing, but the resulting proteins form cis multimers with trans homophilic binding specificity at the multimer level (*Schreiner and Weiner, 2010*; *Thu et al., 2014*; *Wu and Maniatis, 1999*). This incorporation of isoforms into multimers can generate thousands of distinctly homophilic interactors. Intriguingly, the multimer compositions are regulated not only by promoter choice, but also by the relative expression levels of each isoform. PCDHG promotes self-avoidance in cerebellar Purkinje cells and retinal starburst amacrine cells (SACs) (*Lefebvre et al., 2012*). Here again, diversity is required for normal function. When only one isoform was expressed

in SACs, self-avoidance at the single cell level was preserved, but interactions between neighboring SACs were aberrant, resulting in circuit level dysfunction (*Kostadinov and Sanes, 2015*).

Both *Dscam1* and *Pcdhg* promote self-avoidance by conferring self/non-self discrimination. Without extensive isoform diversity, vertebrate DSCAMs are not able to provide this fine level of self-recognition. Our conclusion that DSCAMs function through at least two molecular mechanisms, one that requires a C-terminal PDZ-interacting motif, and one that does not, fits a model in which DSCAMs interact with cell-type-specific adhesion mechanisms to serve their function. This is most easily discussed for self-avoidance, where the deletion of DSCAM or DSCAML1 results in the clumping and fasciculation of homotypic cells – phenotypes that could be described as excessive adhesion. Under this model, the DSCAMs serve a generic role, and the specificity and complexity of the system is conferred by the cell-type-specific adhesion mechanisms. Thus, in cell types in which the PDZ-interacting C-terminus is required, the predominant cell adhesion system may also interact with PDZ-scaffolded complexes. These systems could include Ig-superfamily CAMs such as L1-CAM or NRCAM, or possibly neurexins or neuroligins. Alternatively, the CAMs could interact with PDZ domains indirectly through mediator proteins. In cell types that do not require the C-terminus of DSCAM or DSCAML1, the predominant cell adhesion systems may not involve PDZ interactions, such as cadherins or protocadherins. In cell types with intermediate phenotypes, a mix of PDZ-dependent and –independent mechanisms may be involved. This is not surprising, as each cell type is expected to express more than one class of cell adhesion molecule.

The C-termini of both DSCAM and DSCAML1 interact with at least six different multi-PDZ domain-containing proteins (*Yamagata and Sanes, 2010*) and (*Figure 1—figure supplement 1*). de Andrade and colleagues analyzed the co-localization of DSCAM with seven different multi-PDZ proteins, including MAGI-2 and MAGI-3, and found little co-localization in the adult IPL. Interestingly, there was increased co-localization during development, indicating that these interactions are dynamic and transient (*de Andrade et al., 2014*). None of the analyzed PDZ proteins were clearly localized to specific laminae, but were punctate throughout the IPL. Thus, DSCAM and DSCAML1 could interact with different PDZ proteins in different cell types throughout development.

As mentioned, we have thus far been generally unable to genetically separate functions such as developmental cell death and self-avoidance. Cell types with any phenotype in ΔC mutants had multiple phenotypes, which changed roughly in parallel. For example, DA cells were more numerous with disrupted spacing and neurite fasciculation, while bNOS-positive amacrine cells and Cdh3-GFP RGCs both had a relatively normal density, spacing, and neurite arborization (*Figures 2,3*). In ipRGCs, cell density was variably increased and did not reach statistical significance, and cell spacing and dendrite fasciculation were both intermediate in severity between controls and null mutants (*Figure 3*). This covariation could suggest that there is one primary function and the other phenotypes are secondary to this. Keeley and colleagues tested if cell death was this primary function in DA cells and ipRGCs, and found that while increasing cell density by inhibiting developmental cell death could contribute to spacing defects, it was not sufficient to explain the loss of self-avoidance in *Dscam*$^{-/-}$ mutants (*Keeley et al., 2012*). We confirmed those findings here, and extended them in three ways. (1) We added a third cell type, VGLUT3-positive amacrine cells, and found that their spacing was more disrupted in *Dscaml1*$^{ΔC/ΔC}$ mutants than *Bax*$^{-/-}$ mutants despite having a lower cell density (*Figure 5I–L*). (2) In ipRGCs, cell clustering was more severe in *Dscam*$^{ΔC/ΔC}$ mutants than in *Bax*$^{-/-}$ mutants despite comparable cell densities (*Figure 5A–D*). ipRGC density in *Dscam*$^{-/-}$ mutants is much higher than in *Bax*$^{-/-}$ retinas, complicating the correlation of density and spacing defects. (3) We quantified neurite fasciculation independently from the cell body spacing, finding that cell density can contribute to fasciculation in ipRGCs, but does not significantly drive fasciculation in DA cells or bNOS cells (*Figures 6*, *Figure 6—figure supplement 1*). Thus, promotion of cell death is not the sole primary function of the Dscams. This is consistent with our previous finding that ipRGC clustering and fasciculation occurs in *Dscam*$^{-/-}$ mutants even when cell density is severely reduced by *Pou4f*2 double mutation (*Fuerst et al., 2012*).

As we have not been able to induce clustering and fasciculation without an increase in cell density, it remains possible that the self-avoidance function of Dscams is primary and the clustering and fascicle formation is protective against cell death. It is also possible that the process through which the Dscams prevent fasciculation also promotes proper neurite stratification. We have now identified two examples of aberrant interaction between different cell types. DA cells co-fasciculate with M1 ipRGCs in *Dscam*$^{ΔC/ΔC}$ retinas (*Figure 7E*) and VGLUT3-positive amacrine cells become directly

adjacent to AII amacrine cells in *Dscaml1*$^{\Delta C/\Delta C}$ mutants (*Figure 7P–Q*). As DA and M1 ipRGCs normally co-stratify, this interaction does not result in any misprojection. VGLUT3-positve amacrine neurites, however, are normally confined in the ON-OFF strata between the ChAT layers, not in the ON region proximal to the RGL where AII projections reside. Improper interactions with AII amacrine cells could provide a mechanism for VGLUT3 misprojection (*Figure 7O–Q*).

In summary, DSCAM and DSCAML1 in different cell types function through different molecular mechanisms. We hypothesize that this mechanistic diversity is based on the diverse adhesion systems masked by the DSCAMs. Determining how DSCAMs interact with other systems to mask their adhesivity for self-avoidance, and whether this is the primary function and developmental cell death and synapse maturation are secondary, will require additional studies. However, what these results clearly demonstrate is that the molecular mechanisms through which DSCAMs function will need to be defined on a cell type by cell type basis, and that a single molecular complex or pathway will not account for DSCAMs' function(s) in all cell types.

## Materials and methods

### Mouse strains

All animals were housed in the research animal facility at The Jackson Laboratory under standard housing conditions with a 12:12 light dark cycle and food and water ad libitum. All procedures using animals were performed in accordance with The Guide for the Care and Use of Laboratory Animals and were reviewed and approved by the Animal Care and Use Committee of The Jackson Laboratory. Previously described mouse strains include: *Dscam*$^{-/-}$ = B6.CBy-*Dscam*$^{del17/Rwb}$, RRID:IMSR_JAX:008000, described in (*Fuerst et al., 2008*); *Cdh3-GFP*, RRID:MMRRC_000236-UNC, courtesy of Dr. Andrew Huberman (*Osterhout et al., 2011*); *Bax*$^{-/-}$ = B6.129X1-*Bax*$^{tm1Sjk}$/J, RRID:IMSR_JAX: 002994, described in (*Knudson et al., 1995*); *Dscaml1*$^{-/-}$ = *Dscaml1*$^{GT/GT}$, RRID:MGI:4417834, described in (*Fuerst et al., 2009*). *Dscaml1*$^{-/-}$ mice were genotyped with a previously unreported primer set. A common forward primer (ATGCCACTGTGCCTGGCTGTT) was used with a reverse primer specific to either wild type sequence (CCCAGCAGTTGAGTGCCCTGG) or mutant sequence (TATCCACAACCAACGCACCCAAGC).

### Generation of ∆C mice

To disrupt the PDZ-interacting C-terminus of DSCAM, we replaced the sequence encoding the C-terminal ten amino acids with a Myc epitope tag sequence through standard knock-in techniques. The targeting vector was generated using bacterial recombineering of the BAC BMQ-206A7 (from 129S7/SvEv). We made a recombineering targeting cassette by PCR of a loxP-flanked Neomycin cassette. The forward primer had the following sequence: GTGCAGAGCTGGGACAGGCAGCTAAAA TGAGCAGCTCCCAAGAGTCACTGCTGGACTCCCGGGGCCATTTGAAAGGAAAC**GAACAAAAGC TGATCTCTGAGGAAGATCTGTAA**cggcgcgcctagtcgacttc. This primer was composed of the 60 bases from 90 to 30 bases upstream of the *Dscam* stop codon in exon 33 (underlined), Myc coding sequence with a stop codon (bold), and the 5' end of a loxP-flanked Neomycin cassette (lowercase). The reverse primer had the following sequence: CGGAATTCAGTAAAAAAAAGGTAGCTTTGA TTGGCTCGTTTAAATTGTATTTACAACCGCTGTCCATCAGGTGCCATGTGgcttagtttaaactcgagcc, composed of the 60 bases immediately after the stop codon in exon 33 (underlined) and the 3' end of a loxP-flanked Neomycin cassette (lowercase). The BAC was recombineered in SW102 cells as in (*Warming et al., 2005*). The resulting BAC was digested with PmlI and XbaI to create a 9.2 kb fragment which was subcloned into pSL1180 at EcoRV and XbaI sites. The vector acquired a point mutation resulting in a G to E substitution two residues before the Myc tag. The 9.2 kb targeting vector in pSL1180 was linearized with NotI and electroporated into the C57BL/6N ES cell line JM8. 96 neomycin-resistant clones were screened by long range PCR. Of five positive clones, two were expanded and injected into albino B6(Cg)-*Tyr*$^{c-2J}$/J blastocysts. Chimeric mice were bred to albino B6(Cg)-*Tyr*$^{c-2J}$/J mice to detect germline transmission. To remove the loxP flanked Neomycin cassette, mice harboring the mutation were crossed to a line expressing Cre under the CMV promoter - B6.C-Tg(CMV-cre)1Cgn/J – then backcrossed to C57BL/6J to segregate from Cre. Mice were genotyped by PCR using a primer pair spanning the retained loxP site: CCTCCACCTCTTCCACGCGA-GAAG and AGTAGTCTTTGCGCTGTCTGTGG.

*Dscaml1ΔC* mice were generated in parallel by the same techniques. The BAC RP23-342M16 was recombineered to replace the 30 bases preceding the stop codon in exon 33 with an HA tag and a loxP-flanked Neomycin cassette. The following primers were used: AGGGACTCACTACTCGAAA TGAGCACCCCAGGGGTAGGGCGTTCTCAGAAACAGGGGGCT**TACCCATACGATGTTCCAGATT-ACGCTTAA**cggcgcgccctagtcgacttcg and TGGTGTGCGGGGGCTGGAGGCGCAGAGGTCCCAGTG TGGAGCCCTTCTCCATTTGTCGGCgcttagtttaaactcgagcc, corresponding to exon 33 of *Dscaml1* (underlined), the HA tag (bold), and the loxP-flanked Neomycin cassette (lowercase). The recombineered BAC was digested with BmtI, and the liberated 7.7 kb fragment was subcloned into the pSL1180 vector with a diphtheria toxin negative selection cassette. The resulting targeting vector was linearized by NotI digestion and electroporated into the albino C57BL/6 ES cell line J-A18 (B6 (Cg)-Tyr$^{c-2J}$/J). Clones with confirmed homologous recombination were microinjected into C57BL/6 blastocysts to create chimeric mice. Sperm from chimeric males was genotyped to confirm the presence of the engineered allele, and then used for in vitro fertilization. As with DscamΔC, the resulting mice were crossed with B6.C-Tg(CMV-cre)1Cgn/J to remove the Neomycin cassette, then backcrossed to C57BL/6 to segregate from Cre. Mice were genotyped by PCR using a primer pair spanning the retained loxP site: CCTCCATGAGGAACCTGACTCG and CATGACTGGGGATTTC TTTTTGAC. While the epitope tags were useful for analyzing protein in transfected cells, neither allowed for effective labeling in vivo. This was unfortunate, but not uncommon for a single-copy small epitope tag.

## Yeast two-hybrid

A yeast two-hybrid screen was performed using a Clontech Matchmaker kit as per the manufacturer's instructions (Takara, Mountain View, CA). AH109 yeast transformed with a plasmid encoding the C-terminal 20 amino acids of DSCAM fused to the Gal4 binding domain were mated to Y187 yeast pre-transformed with a mouse cDNA library fused to the Gal4 activation domain. Successful interaction resulted in Gal4-driven expression of HIS3, screened for by survival on plates without histidine and containing 3-AT; ADE2, screened for by survival on plates without adenine; and LacZ, screened for by activity of the β-galactosidase enzyme. Positive interactors, including MAGI-2 and MAGI-3, were verified by immunoprecipitation and Western blot using the Gal4 fusion constructs.

## Expression constructs and transfection

pCAG-Dscam was used to express full length DSCAM (*Schramm et al., 2012*) and pCMV-Dscaml1 to express full length DSCAML1 ([*Yamagata and Sanes, 2008*], Addgene 18738, Cambridge, MA). A Dscam$^{ΔC}$ expression construct mimicking the knock-in mutation was made by PCR from pCAG-Dscam using the primers GGGGACAAGTTTGTACAAAAAAGCAGGCTGGACCATGTGGGATAC TGGCTCTCTCC and GGGGACCACTTTGTACAAGAAAGCTGGGTGTTACAGATCTTCCTCAGAGA TCAGCTTTTGTTCGTTTCCTTTCAAATGGCCCCGGGAG. The PCR product was cloned into the Gateway cloning pDONR201 vector by BP reaction (Thermo Fisher, Waltham, MA) then into pDEST47 by LR reaction (Thermo Fisher) for expression. The Dscaml1$^{ΔC}$ expression vector was made through identical steps starting with pCMV-Dscaml1 using primers GGGGACAAGTTTG TACAAAAAAGCAGGCTGGACCATGTGGCTGGTAACTTTCCTCCTG and GGGGACCACTTTG TACAAGAAAGCTGGGTGTTAAGCGTAATCTGGAACATCGTATGGGTAAGCCCCCTGTTTCTGA-GAACG.

Constructs to express V5-tagged intracellular domains (ICD) were made by PCR from pCAG-Dscam and pCMV-Dscaml1. Both Dscam ICD constructs were made using the forward primer AAT-TAAGAATTC<u>ATGGGTAAGCCTATCCCTAACCCTCTCCTCGGTCTCGATTCTAC</u>*GAGGAGACGGC-GAGAGCAGAGGC* where the V5 tag is underlined and the beginning of the ICD is italicized. The wild type ICD construct was made with the following reverse primer: TTATTCTAGATTATACCAAGG TGTAAGATTTTGC while the Dscam$^{ΔC}$ ICD construct was made with the following reverse primer: TTATTCTAGATTACAGATCCTCTTCTGAGATGAGTTTTTGTTCGTTTCCTTTCAAATGGCCCC. These PCR products were ligated into the pEYFP-N1 vector (Clontech), with the YFP sequence removed, digested at EcoRI (sticky) and NotI (blunted) sites. The Dscaml1 ICD constructs were likewise produced with a common forward primer encoding the V5 tag: AATTAAGAATTC<u>ATGGGTAAGCCTA TCCCTAACCCTCTCCTCGGTCTCGATTCTAC</u>*GCGAAAGAAGAGGAAGGAGAAGAGGC*. Wild type ICD was made with reverse primer: AAATGCGGCCGCCTACACCAGGGTGTAGGATTTGG and

Dscaml1$^{\Delta C}$ with reverse primer TATTGCGGCCGCTTAAGCGTAATCTGGAACATCGTATGGG TAAGCCCCCTGTTTCTGAGAACGC. PCR products were ligated into pEYFP-N1 at EcoRI and NotI sites.

The MAGI-3 expression construct was generated by PCR from cDNA from the neonatal brain using the following primers: CACCATGTCGAAGACGTTGAAGAAG and TCACAGCTGTTTG TCAGCCATG. The PCR product was cloned into pENTR/D-TOPO by TOPO cloning reaction, then into pDEST47 by the LR cloning reaction.

HEK293T cells were obtained from ATCC (Manassas, VA, CRL-11268, RRID:CVCL_1926, lot 62312975) where they were tested free of mycoplasma and their identity was verified by STR analysis. Cells were transfected with Lipofectamine 3000 (Thermo Fisher) according to the manufacturer's protocol.

## Immunoprecipitation and western blot

Immunoprecipitation from whole P0 brain was performed according to standard protocols using mouse anti-DSCAM (1:25, R&D Systems, Minneapolis, MN, clone 36661, RRID:AB_2095452) and protein G magnetic beads (Dynabeads, Thermo Fisher). Western blots were performed according to standard procedures using goat anti-DSCAM primary antibody (1:1000, R&D Systems, RRID:AB_ 2230818). For in vitrostudies, immunoprecpitation was performed 48 hr after transfection with an agarose-conjugated anti-V5 tag antibody (ABCAM, Cambridge, MA, AB1229, RRID:AB_308681) and Western blots with mouse anti-V5 tag (1:2000, Pierce E10/V4RR, RRID:AB_10977225) and rabbit anti-MAGI-3 (1:1000, Sigma, St. Louis, MO, RRID:AB_2619643)

## Immunofluorescence

Tissue preparation and immunofluorescence staining were performed as described previously (*de Andrade et al., 2014*; *Fuerst et al., 2009*). Whole retinas were isolated and fixed in 4% paraformaldehyde for 4–8 hr. Retinas were stained free-floating in 2.5% BSA with 0.5% Triton-x-100 in the indicated antibodies for 48–72 hr at 4°C. After washing off unbound primary antibodies, secondary antibodies were applied in the same solution overnight at 4°C. For sectioning, lenses were removed from enucleated eyes. Eyecups were fixed, then cryopreserved in 30% sucrose and frozen in Tissue-Tek OCT (Sakura, Torrance, CA). Cryosections were cut at 12 µm and immunostained on the slide. Primary antibodies were applied overnight in blocking solution at 4°C, and secondary antibodies for one hour at room temperature. For DSCAM staining, eyecups were only fixed for 50 min on ice. Longer fixation time reduced staining efficiency.

For live staining of HEK293T cells, antibodies were diluted in PBS with 2.5% BSA and applied to cells on ice for 1 hr. Cells were then rinsed in PBS, fixed for 10 min at room temperature in 4% PFA, and counterstained as described above for cryosections.

Histological analysis using H&E was performed using standard staining protocols with a Leica automated slide stainer. Tissue was prepared as above but with paraffin embedding. Sections were cut at 4 µm on a Leica microtome for staining.

## Antibodies

The following antibodies were used at the indicated dilutions: rabbit anti-GFP (1:500, Millipore, Darmstadt, Germany, RRID:AB_91337), mouse anti-DSCAM (1:50, R&D Systems clone 36661, RRID:AB_2095452), sheep anti-tyrosine hydroxylase (1:500, Millipore, RRID:AB_11213126), rabbit anti-bNOS (1:500, Sigma RRID:AB_260796), rabbit anti-melanopsin (1:10,000 gift of Dr. Ignacio Provencio at the University of Virginia or Advanced Targeting Systems, San Diego, CA, RRID:AB_ 1266795), rabbit anti-Dab1 (1:500, Millipore, RRID:AB_2261451), guinea-pig anti-VGLUT3 (1:10,000, Millipore, RRID:AB_2187832), rabbit anti-PKCa (1:1000, Sigma RRID:AB_477345), goat anti-ChAT (1:400, Millipore, RRID:AB_2079751), rabbit anti-HA tag (1:250, Sigma, RRID:AB_260070). All secondary antibodies were alexa-fluor conjugates (1:500, Thermo Fisher ).

## Image analysis
### Cell spacing
X-Y coordinates for each cell body in an image were defined in ImageJ. These coordinates were used to calculate the density recovery profile (DRP), nearest neighbor analysis, and overall cell

density in WinDRP as described previously (*Fuerst et al., 2008*). For each image, the DRPs were normalized so that the overall density was set at 1. This allowed for direct comparison of relative spacing across genotypes independent of the overall cell number. The packing factor (PF) generated by WinDRP was used for statistical comparison (*Rodieck, 1991*). For nearest neighbor analysis, the regularity index (NNRI) was computed by dividing the mean nearest neighbor distance by the standard deviation. The NNRI was also computed for an equal number of randomly distributed points. The ratio of the measured NNRI to random NNRI was reported and used for statistical comparison. The same X-Y coordinates were also used for Voronoi domain analysis in Ka-me (*Khiripet et al., 2012*). Values from 2–4 images per retina were averaged to find a retinal mean. These means were compared across genotype by ANOVA with pairwise Tukey post-hoc tests. In separate experiments where only two genotypes were compared, student's t-tests were employed. Sample sizes were chosen based on previous studies (*Fuerst et al., 2009*, *2008*).

## Elo score

Images were compared for fasciculation of processes using a custom web-based program and an algorithm based on the Elo rating system for ranking chess players (*Elo, 1978*). Users, blind to genotype, were instructed to choose which of two images was the least fasciculated (i.e., most like the wild type). Two random images were presented, and the user clicked on his or her choice. After choosing, a new pair of images was presented. All images began with a score of 0, and after each matchup their scores were updated. The winning image gained points while the other image lost the same number of points. The number of points exchanged was based on the scores of each image before the matchup. As a result, in an upset scenario (i.e., an image with a low score 'beat' an image with a higher score) more points were exchanged than if the originally higher scored image had won. This allows a group of images to sort into their true ranking without performing every head-to-head matchup. The trial was considered to have reached an endpoint when further matchups did not change the relative genotype rankings. Each image set was scored by 6–8 users. Scores from each individual retina were averaged (1–4 images per retina) and these retinal mean scores were used to compare across genotypes using a pairwise Wilcoxon ranked sum test.

## VGLUT3 projections

VGLUT3 projections were analyzed in Imaris software (Bitplane). High-resolution confocal stacks of Dab1 and VGLUT3 co-labeling were reconstructed in 3D. Blind to genotype and to the VGLUT3 channel, the stack was cropped to include only the Dab1-positive AII terminals, then projections were made through both channels. The VGLUT3 projections were thresholded at three levels in ImageJ and the percent area occupied was calculated. Values from two images per retina were averaged and this mean score was compared across genotypes and threshold level by two-way ANOVA.

## DSCAM protein levels

DSCAM protein expression was quantified in cryosections by two methods. Single plane confocal images were collected from sections stained for DSCAM and melanopsin using identical imaging parameters including laser power, pinhole size, and PMT gain and offset. Both channels were thresholded in Adobe Photoshop, again using identical parameter for all images. The overlap between the thresholded DSCAM and melanopsin channels was extracted using Photoshop's multiply function. The ratio of the overlap area to the melanopsin area – both quantified in NIH ImageJ – was calculated. The second method allowed quantification independent of thresholding. The intensity of DSCAM fluorescence was measured along a one-dimensional line 10 µm long projecting into the IPL directly adjacent and perpendicular to the INL. The average fluorescent intensity along the line was calculated. For both methods, multiple measurements were averaged per retina, and these retinal means were compared across genotypes by ANOVA with pairwise Tukey post-hoc tests.

## Acknowledgements

We would like to thank the Scientific Services at the Jackson Laboratory for many aspects of this project, including the Cell Biology and Microinjection Services for assistance in generating mutant strains of mice, and the Histology and Microscopy services for sample preparation. We would also

like to thank Dave Walton and Keith Shepard for their assistance in designing the Elo image quantification system, and Dr. Joshua Weiner for helpful discussion of the manuscript.

## Additional information

### Funding

| Funder | Grant reference number | Author |
| --- | --- | --- |
| National Eye Institute | F32EY021942 | Andrew M Garrett |
| National Institute of Neurological Disorders and Stroke | T32NS051112 | Andrew M Garrett |
| National Eye Institute | F32EY022825 | Abigail LD Tadenev |
| National Eye Institute | RO1EY018605 | Robert W Burgess |
| National Institute of Neurological Disorders and Stroke | RO1NS054154 | Robert W Burgess |
| National Institute of Neurological Disorders and Stroke | R21NS090030 | Robert W Burgess |

The funders had no role in study design, data collection and interpretation, or the decision to submit the work for publication.

### Author contributions

AMG, Experiments were conceived, Executed all genetic analyses, collected and analyzed data on retinal phenotypes, wrote the manuscript with input from all authors; ALDT, Acquisition of data, Drafting or revising the article, Performed immunoprecipitations and western blots; YTH, Acquisition of data, Analysis and interpretation of data, Drafting or revising the article, Identified PDZ interactions; PGF, Conception and design, Acquisition of data, Identified PDZ interactions, Generated targeting vectors for C-terminal deletion of DSCAM; RWB, Conception and design, Analysis and interpretation of data, Drafting or revising the article, Experiments were conceived, wrote the manuscript with input from all authors

### Author ORCIDs

Robert W Burgess, http://orcid.org/0000-0002-9229-3407

### Ethics

Animal experimentation: All procedures using animals were performed in accordance with The Guide for the Care and Use of Laboratory Animals and were reviewed and approved under Animal Use Summary 1026 by the Animal Care and Use Committee of The Jackson Laboratory.

## Additional files

### Supplementary files

• Supplementary file 1. p-values from all pairwise comparisons. p-values were calculated from each pairwise comparison across genotypes using the indicated measurement and test. NN = nearest neighbor analysis; PF = packing factor derived from DRP (density recovery profiling).

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
