## [Decision Letter]

Thank you for submitting your article "Differential dependence on the PDZ-interacting C-termini of DSCAM and DSCAML1 in different cell types" for consideration by *eLife*. Your article has been reviewed by two peer reviewers, and the evaluation has been overseen by a Reviewing Editor and a Senior Editor. The reviewers have opted to remain anonymous.

The reviewers have discussed the reviews with one another and the Reviewing Editor has drafted this decision to help you prepare a revised submission.

Garrett et al. address an interesting question regarding the role of the PDZ interacting domains of DSCAML1 and DSCAM. By replacing the C-terminal 10 amino acids with an epitope, and knocking these alleles into the endogenous loci, they address the requirement for these terminal amino acids. Their findings are somewhat surprising, as they show differential dependence of these domains among several retinal cell types. As indicated by the Reviewers, their conclusions need additional data, as well as a more measured interpretation. The major difficulty in interpretation is pointed out by Reviewer 1, which derives from the design of their knock-in allele. By using only one allele, that of a deletion with an epitope substitution, they have not ruled out that the phenotype is due to the epitope. This ambiguity needs to be acknowledged and discussed. Further, in light of the caveats in interpretation, the authors should consider modifying their title. In addition, as both reviewers point out, there is not enough quantification and resolution of the level and location of the knock-in protein within the retina. Given the differential response of different cell types, this is an important point. The suggestions of Reviewer 2 regarding more accurate measurements of protein and localization need be carried out. In addition, the other suggestions of Reviewer 2 should be carried out. The caveats in the interpretation pointed out by Reviewer 1, as well as a more complete discussion of previous work of DsCAMs in *Drosophila*, and the model for protocadherins in self-avoidance in mouse, should also be included.

The entirety of the reviews are appended below.

Reviewer #1:

Fuerst and Burgess and their co-workers have described in detail in previous studies phenotypes in associated with loss of function alleles of DSCAM and DSCAML1. They have shown that DSCAM regulates self-avoidance, cell death and synapse formation. To begin to dissect these multiple functions they use knock-in technology to construct mutations in the cytoplasmic tails in which the C-terminal PDZ-binding domain was replaced with an epitope tag. The assumption was that different functions would be mediated by different signaling pathways downstream the cytoplasmic domains. That is, the lack of the PDZ domain would abrogate one function, perhaps cell death, and leave the others (e.g. synapse formation) intact. To their surprise, they discovered that rather than disrupting one pathway or another the loss of the PDZ-binding site in both proteins led to selective disruption of function in different cell types. The authors argue, based on these findings, that there are different signaling requirements in different cells for the same function.

This is a reasonable interpretation of the data. But it is important to recognize the limitations of the analysis and it is important that these issues be raised and considered. That is the experiments presented are very good but they are limited and this should be acknowledged in the text. This is *eLife* so we expect frankness and not a vanity journal where the editors expect a crisp and important message, even if this is not warranted by the data.

First, it is impossible for the authors to know whether the phenotypes they observe are due to the insertion of the tag, the loss of the PDZ binding site or both. A more careful control would have been to insert the tag towards the C-terminal with PDZ domain intact. If this rescues, the authors could include the same tag with a mutated PDZ domain binding site. If we were playing hardball, this would negate their data entirely. But it is a high bar and I think the paper has value and should be published.

Second, the authors argue that expression is normal, including the levels and distribution of the protein. The western blots were from the neonatal brains but not from retina, the tissue in question. There is no control for loading presented. The expression is assessed in 3 week old mice; the authors should address this expression in the context of what is known about the developmental etiology of the phenotypes. Is this the relevant stage for the defects seen in nulls or is it considerably later?

Third, protein distribution is only considered in a very gross way and indeed what one would wish to assess is whether the level and distribution in the neurons in which phenotypes were assessed were normal. To their credit they attempt this in Figure 1 and the accompanying supplementary figure. But the data are very low resolution and very low quality making it virtually impossible to make any claims about normal localization of DSCAM in any particular cell type. There are methods to assess expression in specific cell types but this would have required much more sophisticated genetic engineering in the knock in alleles they generated. I am not suggesting they should have done this or that they should go back at to it now. I think it is critical that the authors indicate that they cannot adequately assess whether the phenotypes reflect differences in expression and/or localization in different cell types rather than arising from their preferred explanation that this is due to differences in molecular function in different neuronal cell types.

And finally, the authors provide one explanation for differences in different cell types in which DSCAM or DSCAML1 associates with adhesion molecules and thereby prevents fasciculation between processes of the same cell. They state in the Discussion that:

"These results and conclusions fit a model in which DSCAMs interact withcell-type-specific adhesion mechanisms to serve their function. […] In cell types with intermediate phenotypes, a mix of PDZ-dependent and -independent mechanisms may be involved. This is not surprising, as each cell type is expected to express more than one class of cell adhesion molecule."

This paragraph will not be readily appreciated by most readers and some additional information would make it more comprehensible and thus interesting to them. The statement of lack of diversity comes out of nowhere; this paragraph is taken from the Discussion and there has up until this point not been any reference to the fly Dscam diversity or to how it regulates self-avoidance. This would not be necessary (arguably a little odd but not necessary) if they were not to contrast this in the context of interpreting their rescue data. The mechanism by which DSCAMs in mammalian systems (i.e. the mouse) are proposed to work is very different from the description in flies and very different from the mechanisms more recently described for clustered protocadherins in the mouse. The authors should be explicit about this. That is describing the fly model (and protocadherin model-largely the same) and then contrast this with the model for DSCAM's role in self-avoidance in the mouse retina. It is a particularly interesting model and entirely plausible. I think putting their model for DSCAM and DSCAML1 in the broader context of self-avoidance and then interpreting it in terms of differential association with different adhesion molecules (i.e. some association via PDZ and others through other mechanisms) is intriguing.

Reviewer #2:

This is an interesting article exploring the molecular interactions that are required for Dscam and Dscaml to regulate cell spacing and neurite organization of retinal neurons during development. The major finding is that the PDZ-interacting C-terminal motif of Dscam and Dscaml is important in some but not all retinal cell types in executing the proper function of these adhesion molecules. The authors convincingly show a differential requirement for the PDZ-binding motif in a subset of retinal neurons by examining cell density, mosaic organization and dendrite fasciculation of several amacrine and ganglion cell types. Quantification of these parameters are provided and by and large, the figures support the major conclusions. For some conclusions, the authors still need to provide quantitative data to support their statements (these are listed in the minor comments). The major finding is somewhat unexpected and thus the study does represent a significant addition to current knowledge of how Dscam and Dscaml proteins work.

---

## [Author Response]

*Garrett et al. address an interesting question regarding the role of the PDZ interacting domains of DSCAML1 and DSCAM. By replacing the C-terminal 10 amino acids with an epitope, and knocking these alleles into the endogenous loci, they address the requirement for these terminal amino acids. Their findings are somewhat surprising, as they show differential dependence of these domain among several retinal cell types. As indicated by the Reviewers, their conclusions need additional data, as well as a more measured interpretation. The major difficulty in interpretation is pointed out by Reviewer 1, which derives from the design of their knock-in allele. By using only one allele, that of a deletion with an epitope substitution, they have not ruled out that the phenotype is due to the epitope. This ambiguity needs to be acknowledged and discussed. Further, in light of the caveats in interpretation, the authors should consider modifying their title. In addition, as both reviewers point out, there is not enough quantification and resolution of the level and location of the knock-in protein within the retina. Given the differential response of different cell types, this is an important point. The suggestions of Reviewer 2 regarding more accurate measurements of protein and localization need be carried out. In addition, the other suggestions of Reviewer 2 should be carried out. The caveats in the interpretation pointed out by Reviewer 1, as well as a more complete discussion of previous work of DsCAMs in Drosophila, and the model for protocadherins in self-avoidance in mouse, should also be included.*

We would like to thank the reviewers for their generally positive response and constructive suggestions, as well as their acknowledgement that our study addresses an interesting and significant question. We feel we have addressed the concerns that were raised by including additional data and details, and by editing the manuscript as requested. Details of these changes are provided below. We have also changed the title as suggested in the editorial comments.

*The entirety of the reviews is appended below.*

*Reviewer #1:*

*Fuerst and Burgess and their co-workers have described in detail in previous studies phenotypes in associated with loss of function alleles of DSCAM and DSCAML1. They have shown that DSCAM regulates self-avoidance, cell death and synapse formation. To begin to dissect these multiple functions they use knock-in technology to construct mutations in the cytoplasmic tails in which the C-terminal PDZ-binding domain was replaced with an epitope tag. The assumption was that different functions would be mediated by different signaling pathways downstream the cytoplasmic domains. That is, the lack of the PDZ domain would abrogate one function, perhaps cell death, and leave the others (e.g. synapse formation) intact. To their surprise, they discovered that rather than disrupting one pathway or another the loss of the PDZ-binding site in both proteins led to selective disruption of function in different cell types. The authors argue, based on these findings, that there are different signaling requirements in different cells for the same function.*

*This is a reasonable interpretation of the data. But it is important to recognize the limitations of the analysis and it is important that these issues be raised and considered. That is the experiments presented are very good but they are limited and this should be acknowledged in the text. This is eLife so we expect frankness and not a vanity journal where the editors expect a crisp and important message, even if this is not warranted by the data.*

*First, it is impossible for the authors to know whether the phenotypes they observe are due to the insertion of the tag, the loss of the PDZ binding site or both. A more careful control would have been to insert the tag towards the C-terminal with PDZ domain intact. If this rescues, the authors could include the same tag with a mutated PDZ domain-binding site. If we were playing hardball, this would negate their data entirely. But it is a high bar and I think the paper has value and should be published.*

We have acknowledged this important caveat in the Results (subsection “Differential dependence DSCAM/PDZ interactions across cell types”) and Discussion (Second paragraph). In addition, we have verified that the tagged ∆C proteins disrupt interactions with MAGI-3. These experiments are in Figure 1 and Figure 4—figure supplement 1 and are described in the Methods section. While these experiments do not rule out possible interfering activities of the tags, they do support our contention that we have disrupted PDZ-domain interactions.

*Second, the authors argue that expression is normal, including the levels and distribution of the protein. The western blots were from the neonatal brains but not from retina, the tissue in question. There is no control for loading presented. The expression is assessed in 3-week-old mice; the authors should address this expression in the context of what is known about the developmental etiology of the phenotypes. Is this the relevant stage for the defects seen in nulls or is it considerably later?*

We have added expression and localization analysis at P7 and P9. See below.

*Third, protein distribution is only considered in a very gross way and indeed what one would wish to assess is whether the level and distribution in the neurons in which phenotypes were assessed were normal. To their credit they attempt this in Figure 1 and the accompanying supplementary figure. But the data are very low resolution and very low quality making it virtually impossible to make any claims about normal localization of DSCAM in any particular cell type. There are methods to assess expression in specific cell types but this would have required much more sophisticated genetic engineering in the knock in alleles they generated. I am not suggesting they should have done this or that they should go back at to it now. I think it is critical that the authors indicate that they cannot adequately assess whether the phenotypes reflect differences in expression and/or localization in different cell types rather than arising from their preferred explanation that this is due to differences in molecular function in different neuronal cell types.*

We have added the experiment suggested by Reviewer 2 to quantify DSCAM localization with a specific affected cell type (ipRGCs, Figure 1, Figure 1—figure supplement 2) and have acknowledged that due to the limits of resolution in these experiments, we cannot definitively rule out cell-type specific changes in protein stability or localization in the Results (subsection “Differential dependence DSCAM/PDZ interactions across cell types” and in the Discussion (second paragraph).

*And finally, the authors provide one explanation for differences in different cell types in which DSCAM or DSCAML1 associates with adhesion molecules and thereby prevents fasciculation between processes of the same cell. They state in the Discussion that:*

*"These results and conclusions fit a model in which DSCAMs interact withcell-type-specific adhesion mechanisms to serve their function. […] This is not surprising, as each cell type is expected to express more than one class of cell adhesion molecule."*

*This paragraph will not be readily appreciated by most readers and some additional information would make it more comprehensible and thus interesting to them. The statement of lack of diversity comes out of nowhere; this paragraph is taken from the Discussion and there has up until this point not been any reference to the fly Dscam diversity or to how it regulates self-avoidance. This would not be necessary (arguably a little odd but not necessary) if they were not to contrast this in the context of interpreting their rescue data. The mechanism by which DSCAMs in mammalian systems (i.e. the mouse) are proposed to work is very different from the description in flies and very different from the mechanisms more recently described for clustered protocadherins in the mouse. The authors should be explicit about this. That is describing the fly model (and protocadherin model-largely the same) and then contrast this with the model for DSCAM's role in self-avoidance in the mouse retina. It is a particularly interesting model and entirely plausible. I think putting their model for DSCAM and DSCAML1 in the broader context of self-avoidance and then interpreting it in terms of differential association with different adhesion molecules (i.e. some association via PDZ and others through other mechanisms) is intriguing.*

Thank you, we agree that this interpretation is clarified by added discussion of other instances of self-avoidance. We have added text to the Introduction mentioning *Drosophila* Dscams (second paragraph) and further describing previous findings with *Drosophila* Dscam1/mouse Pcdhg model in the Discussion (third and fourth paragraph).